# Single-neuron spiking variability in hippocampus dynamically tracks sensory content during memory formation in humans

Leonhard Waschke[1,2,10], Fabian Kamp[1,2,3,10], Evi van den Elzen [4], Suresh Krishna[5], Ulman Lindenberger [1,2], Ueli Rutishauser [6,7,8,9] & Douglas D. Garrett [1,2] ✉

During memory formation, the hippocampus is presumed to represent the content of stimuli, but how it does so is unknown. Using computational modelling and human single-neuron recordings, we show that the more precisely hippocampal spiking variability tracks the composite features of each individual stimulus, the better those stimuli are later remembered. We propose that moment-to-moment spiking variability may provide a new window into how the hippocampus constructs memories from the building blocks of our sensory world.

Prior to memory formation, visual stimulus properties are encoded along the ventral visual pathway[1,2]. Neural selectivity is thought to shift from low-level image properties towards more composite features as information propagates from visual cortex to the medial temporal lobe (MTL)[3]. In the hippocampus, a high-level endpoint along this pathway, a variety of visual features are believed to be transformed into coherent representations during memory encoding[4–6], but what granularity of information can be conjoined by the hippocampus is not clear.

Considering the behavioural relevance of basic sensory encoding in cortex[7,8], it is plausible that sensory features also need to be directly accessible to the hippocampus to enable the formation of conjunctive representations[9,10]. The tracking of simple visual features in the hippocampus may thus play a crucial role to generate rich and detailed memory traces[5,6,10]. However, since there are minimal direct visual-cortical afferents to hippocampal areas[1,3], it is also plausible that the hippocampus preferably tracks more composite features derived from a mixture of simpler features[3]. Classical work on single cells[11] as well as

more recent work on neural populations[12] suggests that hippocampal activity primarily represents abstract knowledge. Nonetheless, a direct comparison of the types of visual building blocks required for hippocampal memory encoding has not yet been made. Particularly, their respective relevance for individual differences in memory performance is unknown.

One primary challenge in probing this fundamental question is that tailored experimental approaches that can separate simple and composite sensory features have not been leveraged. We argue that the architecture of multi-layer computational vision models can be used to differentiate between simple and composite visual features of any stimulus a participant may encode. Specifically, feature maps from early layers of these models mark the presence of simple features in the image, while feature maps in later layers resulting from many non-linear combinations of the previous layers mark the presence of more complex, composite features[13]. Hierarchical vision models thus offer a unified framework to characterize both low- and high-level visual

[1]Max Planck UCL Centre for Computational Psychiatry and Ageing Research, Max Planck Institute for Human Development, Berlin, Germany. [2]Center for Lifespan Psychology, Max Planck Institute for Human Development, Berlin, Germany. [3]Max Planck School of Cognition, Max Planck Institute for Human Cognitive and Brain Sciences, Leipzig, Germany. [4]Tilburg School of Social and Behavioral Sciences, Tilburg University, Tilburg, The Netherlands. [5]Department of Physiology, McGill University, Montreal, Canada. [6]Department of Neurosurgery, Cedars-Sinai Medical Center, Los Angeles, CA, USA. [7]Department of Neurology, Cedars-Sinai Medical Center, Los Angeles, CA, USA. [8]Division of Biology and Bioengineering, California Institute of Technology, Pasadena, CA, USA. [9]Center for Neural Science and Medicine, Department of Biomedical Sciences, Cedars-Sinai Medical Center, Los Angeles, CA, USA. [10]These authors contributed equally: Leonhard Waschke, Fabian Kamp. ✉e-mail: garrett@mpib-berlin.mpg.de

features, which may be leveraged by the hippocampus during memory formation. But what signature of neural activity could track different visual features in the hippocampus?

In recent years, moment-to-moment variability of neural activity has emerged as a behaviourally relevant measure offering substantial insights into brain function beyond conventional approaches such as average brain activity[14]. Indeed, single-neuron spiking rate and variability may support distinct functions in the visual system, as various features of the visual input can affect neural variability independently from the mean activity[15,16]. For instance, computational work suggests that uncertainty with regards to visual input may be uniquely linked to variability in the spike train[16]. Furthermore, a recent study showed that depending on the perceptual statistics of visual input neurons in V1 of macaques adjust their spiking dynamics[16,17], leading to trial-by-trial changes in neural variability. Remarkably, adaptive changes of neural variability can also account for individual differences in behaviour as evidence from fMRI showed that individuals who exhibit increased visual-cortical BOLD variability in response to more feature-rich stimuli also demonstrate superior cognitive performance[18].

However, it is unknown whether these findings also extend to neural activity in the hippocampus: do features of the visual input relate to hippocampal spiking variability? Moreover, it is an open question if the adaptive regulation of neural variability is relevant for memory encoding. Here, we propose the use of partial least squares correlation (PLS)[19] to estimate, for each subject, the relation between (a) trial-by-trial fluctuations of spiking variability in the hippocampus and (b) multivariate image features. We posit that individuals who display a stronger trial-by-trial association between hippocampal variability and the content of visual input should also exhibit better memory, providing evidence that visual features have successfully been encoded by the hippocampus during memory formation.

## Results

Here, we analysed single-neuron hippocampal recordings from 34 human patients (Fig. 1a) during a visual encoding and recognition memory task. All neural analysis was done on the single-subject level based on simultaneously recorded neurons ($12 \pm 11$ hippocampal neurons per individual and session, total $N = 411$). We only included high-quality, well-isolated units that satisfied all spike sorting quality metrics[20]. The variability of single-neuron hippocampal activity was estimated on every trial during the encoding part of the task using the permutation entropy (PE) of the spike trains[21].

To estimate simple and composite visual features from the images presented at encoding, we employed two computational vision models, HMAX and VGG16 (Fig. 1b)[13,22]. The hierarchical structure of these models allows the differential analysis of simple and composite visual features by means of their layer-wise feature maps (i.e. heat maps, Figs. 1b and S1). Feature maps of early layers capture the presence of simpler visual features in the image. These simpler features are then aggregated non-linearly across the model layers, resulting in more complex, composite features in the late layers. Importantly, we used estimates from HMAX and VGG16 to counterbalance their respective limitations: HMAX is a biologically inspired model of early visual processing (V1-V4)[13,23] but is limited in its ability to extract higher-level visual features. Conversely, although not biologically inspired per se, VGG16 is one of the most employed computer vision models in the field; it comprises many more layers than HMAX and is thus better suited for the analysis of more aggregated, higher-level features[22]. Based on the feature maps for each image, we calculated three summary metrics: the spatial sum, standard deviation (SD), and number of non-zero entries across all values in the map. The number of features varies by layer depth, with later layers containing more features. Thus, to ensure comparability across layers, we used PCA to extract the first principal component scores separately for each model layer and image metric (yielding one score per layer and image metric for each image),

which we used for further analysis (see Methods for details). We then estimated the relationship between these image feature metrics and hippocampal spike variability using within-participant latent modelling[19] (Fig. 1d). Finally, we tested whether stronger coupling between visual features and hippocampus spike modulation reflects better memory performance, and examined whether simple or composite visual features are most crucial in this context.

### Trial-level coupling between spike entropy and layer-wise image feature metrics during encoding

We first addressed the individual trial-level coupling between encoding spike entropy and layer-wise image feature metrics via partial least squares (PLS) (see Fig. 1d and Methods for details). Figure 2a depicts within- and across-layer stimulus weights for each subject, highlighting wide individual differences in the relative importance of image features in coupling to hippocampal spike variability. We also revealed individual correlations of hippocampal spike entropy for early as well as late-layer features in individual subjects (Fig. 2b). For detailed patterns of model-wise feature weights, please see Supplements (Fig S2). Individual late-layer coupling estimates were higher than early-layer coupling estimates, and the two were substantially correlated (rho = 0.59, $p_{permuted} = 0.0002$; Fig. 2c). Hence, the entropy of trial-wise hippocampus spike entropy during encoding was coupled to simple as well as composite features of images presented during encoding, with stronger coupling to more composite (late-layer) features.

### Testing the link between hippocampal spike PE-to-image feature coupling and memory performance

We then tested the relevance of individual spike-to-image feature coupling at encoding to later recognition memory performance (where performance = principal component score capturing various measures of accuracy (mean = 0.73), dprime (mean = 1.4), and confidence (mean = 2.5 out of 3; see Fig. 3 and Methods)). We also contrasted the predictive power of hippocampal spike PE-to-image feature coupling using early, late, or all computational vision model layers. In this way, we sought to unfold their relative importance for the translation of visual features into reliable memory traces during encoding. Later memory performance was positively (but not significantly) correlated with the coupling of hippocampal spike variability during encoding to image features for early-layers (Fig. 3a, $\eta = 0.32$, $p_{permuted} = 0.094$) and significantly for all layers ($\eta = 0.42$, $p_{permuted} = 0.019$), but most strongly so for late-layers ($\eta = 0.54$, $p_{permuted} = 0.001$). $P$ values were computed using non-parametric permutation tests where the $p$ value was defined as the proportion of permutations revealing a higher eta estimate than the estimate from the original, unpermuted data (see Online Methods). See Fig. S3 for behavioural variable-specific distributions and model results. Additionally, Fig. S4 illustrates that memory performance and accuracy of the animacy judgement task at encoding were not correlated, thus, ruling out that low-performing patients were simply less engaged during memory encoding.

Importantly, the relationship between spike variability coupling and memory performance remained qualitatively unchanged for late layers after controlling for the coupling of hippocampal spike variability to early layers (Fig. 3b, $\eta_{partial} = 0.45$, $p_{permuted} = 0.016$), and all layers ($\eta_{partial} = 0.38$, $p_{permuted} = 0.045$). In reverse, early-layer coupling ($\eta_{partial} = -0.02$, $p_{permuted} = 0.625$) and all-layer coupling ($\eta_{partial} = -0.11$, $p_{permuted} = 0.77$) did not predict memory performance once controlled for late-layer coupling.

To ensure that trial-wise spike rate did not confound the behavioural effect of the coupling between spike PE and late-layer image features, we repeated the subject-level PLS analysis using spike rate as a target measure of neural activity instead of spike PE. This yielded subject-level estimates of the latent correlations between spike rate and late-layer features, which were then used as a control. As expected, coupling between hippocampal spike PE and late-layer image features

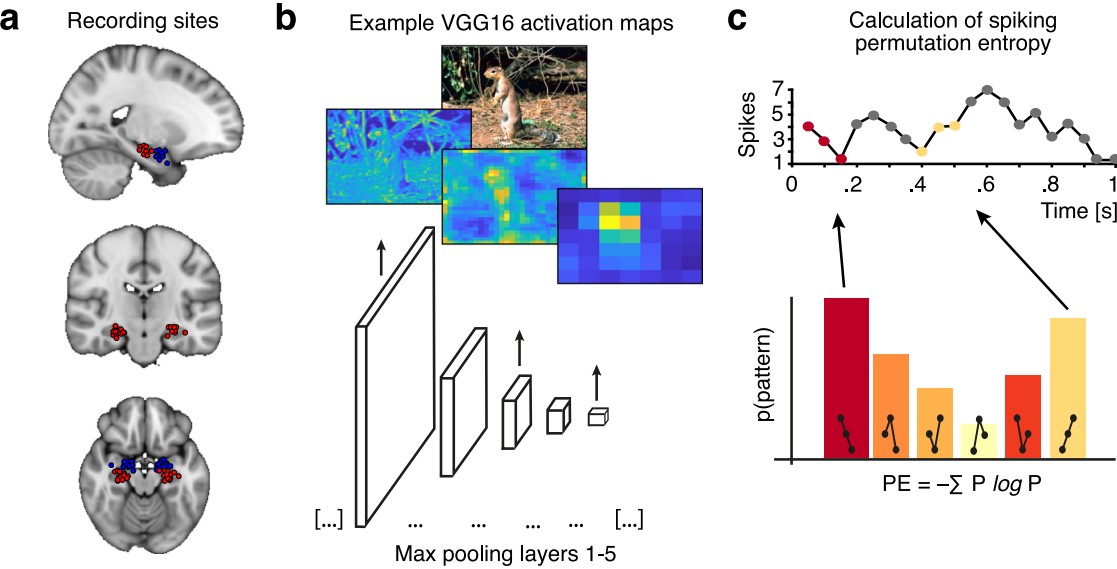

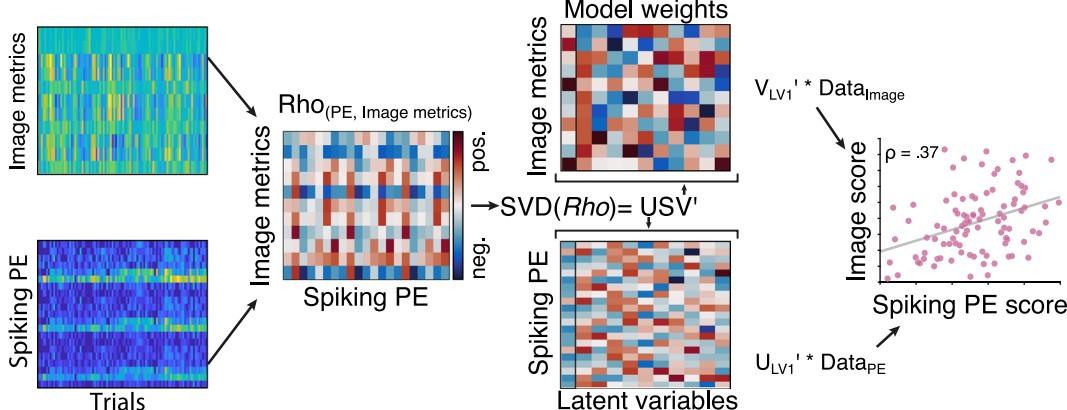

**Fig. 1 | Estimating latent coupling between image features and hippocampal spike entropy (PE). a** Recording sites of depth electrodes for all participants with available probe coordinates. Hippocampus sites in red, amygdala sites in blue (top: $x = -21$, middle: $y = -19$, bottom: $z = -17$). **b**: VGG16 (trained on Imagenet) was used to predict activation maps at five layers of varying depth (max pooling layers 1–5) for images previously shown to participants at encoding, resulting in feature-wise activation maps. The mean across layer-wise features is shown for two example images and max pooling layers 1, 3 and 5. We extracted three summary metrics per layer and feature (sum, standard deviation, number of non-zero elements) before subjecting each layer-wise summary matrix (# images * # features) to a principal component analysis (PCA). In all further analyses, we relied on the first component score of each image, layer, and summary metric. Example image used here from[20] (https://creativecommons.org/licenses/by/4.0/). **c**: Spike entropy was calculated per neuron and trial based on the first second of image encoding (for all neurons with PE > 0.0001 and trials with >1/3 of neurons spiking). In brief, permutation entropy works by transforming signals into patterns (here: length = 3) and counting these patterns before calculating the Shannon entropy of the pattern distribution. **d**: Within-person correlations were computed by decomposing the rank-correlation matrix of trial-wise spike PE (per neuron) and image feature metrics using partial least squares (PLS). Singular value decomposition (SVD) of the rank-correlation matrix results in neural and stimulus weights per latent variable (LV). The weights of the first LV (first column outlined in black) were used to reduce the dimensionality of neural and feature matrices into scores for each trial. The rank correlation between both weighted variables represents the latent estimate of across-trial coupling between image features and hippocampus spike PE (right-most panel). Panels in (**d**) show data from a single subject in our sample.

predicted memory formation success over and above coupling between standard spike rate and late-layer features (Fig. 3b; $\eta_{partial} = 0.39$, $p_{permuted} = 0.022$). Furthermore, controlling for an additional set of potential confounds (number of trials and neurons, task variant, duration of encoding, and participant age) did not reduce the link between spike PE modulations and memory performance (Fig. 3b; $\eta_{partial} = .64$, $p_{permuted} = 0.00$).

Finally, given the presence of memory- and visually-sensitive neurons in the amygdala[20,24], we repeated all analyses above for neurons recorded in the amygdala within the same group of patients ($17 \pm 10$ amygdala neurons per individual and session, total $N = 507$). Unlike for hippocampal neurons, the coupling of amygdala spike

entropy to visual features was not significantly predictive of memory performance (Fig. 3c; all $p_{permuted} > 0.6$). Also, controlling for amygdala coupling had minimal impact on the coupling between hippocampal PE and late-layer image features (Fig. 3c, $\eta_{partial} = 0.57$, $p_{permuted} = 0.01$). These results speak to the anatomical specificity of the coupling between visual features and neural dynamics that might trace back to the extraction and conjunction of composite information achieved by a diverse set of neurons in the hippocampus specifically.

## Discussion
Collectively, these results represent first evidence that intra-individual coupling between hippocampal spiking variability and image features

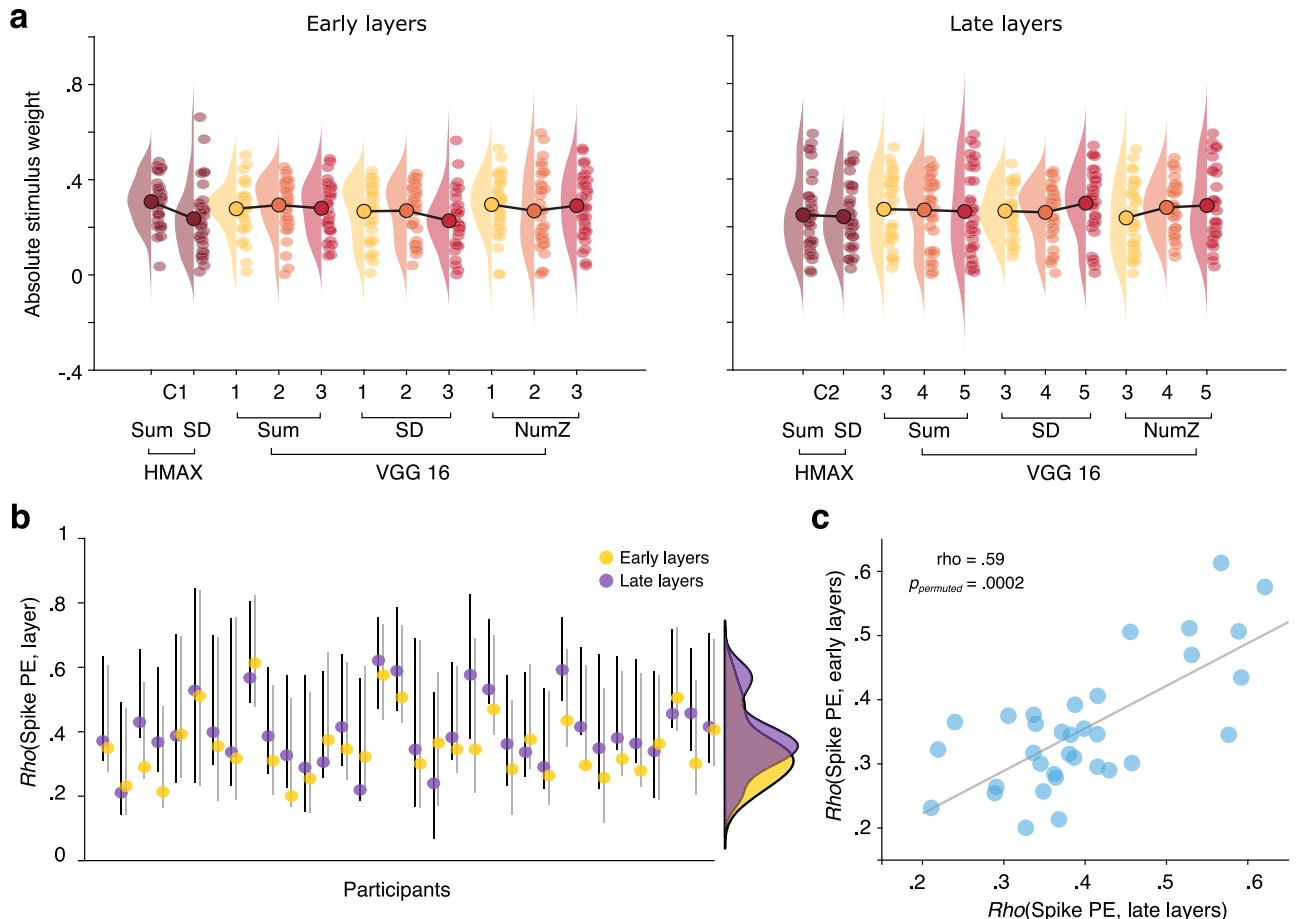

**Fig. 2 | Coupling of hippocampus spike entropy to image features. a**: Feature- and layer-wise absolute stimulus weights for early layer (left) and late layer models (right), both based on the same hippocampal spiking data. Raincloud plots[52] contain participant-wise estimates (single dots), densities, and grand averages (circled big dots, connected by black lines) grouped by vision models (HMAX, VGG16) and feature metrics (Sum, SD, number of non-zero entries (NumZ)). **b**: Absolute latent correlation estimates (Spearman) including 95% bootstrapped confidence intervals resulting from individual PLS models estimating hippocampal spike PE coupling to late (purple) and early-layer image features (yellow). Each dot represents one participant. **c**: Positive between-subject correlation between hippocampal spike PE coupling to late and early-layer visual features, illustrating stable individual coupling to image features overall ($p_{permuted} = 0.0002$, $n = 34$). Dots represent participants.

during encoding is crucial for the successful formation of memories. Importantly, within individuals, hippocampal spike entropy was coupled more strongly to composite than to simple sensory features (Fig. 2b), and this late-layer hippocampal spike coupling dominantly predicted memory performance up to 30 min later (Fig. 3).

Our results support the idea of a representational hierarchy along the ventral visual stream in which simple visual features are progressively aggregated into more composite features from visual cortex to the hippocampus[2,10,25] (with some evidence also coming from fMRI-based studies[8,10,26,27]). However, until now, a direct comparison of the relative importance of early- (simple) and late-layer (composite) features for memory encoding has remained absent from the literature. By using hierarchical vision models, we provide evidence that the dynamic encoding of composite features in the hippocampus (located at the end of the ventral stream) is substantially more relevant to memory performance than the encoding of simpler features. Thus, our findings provide robust evidence for the behavioural importance of the ventral visual representational hierarchy, directly through a crucial new lens of hippocampal neural variability. In particular, individuals showing higher adaptive modulation of spiking variability by late-layer visual features at encoding also performed better during subsequent memory retrieval.

Our findings align well with existing literature on single cell and population coding in the human hippocampus showing that hippocampal activity tracks abstract concepts and knowledge[11,12]. In our case, however, we analysed the visual input continuously with respect to properties of early- and late-layer feature maps instead of using contrasts between object categories. Our latent modelling approach permitted us to leverage trial-to-trial variations of different visual input features independently of semantic image categories, a flexible approach for estimating the encodable content of stimuli without making assumptions about what features might be relevant for encoding the images into memory.

Our results also provide support for the idea that the hippocampus might be able to generate conjunct information by combining sensory, object, and relational aspects into a rich and generalizable memory trace[2,5]. The absence of memory-relevant spike variability coupling in the amygdala, despite this structure's known role in memory formation[24] and direct afferents from visual cortical areas[28], further highlights the unique role of the hippocampus as a dynamic conjunction hub of more aggregated visual input during memory formation. This dissociation of hippocampus and amygdala dynamics is further supported by recent findings showing that item-specific memory signals may only be present in the hippocampus, while more generic memory signals may also be present in the amygdala and other brain areas[29]. It is feasible that our results provide a basis/mechanism for understanding how item specificity may exist in hippocampus; by modulating neural variability (PE) more tightly in line with the

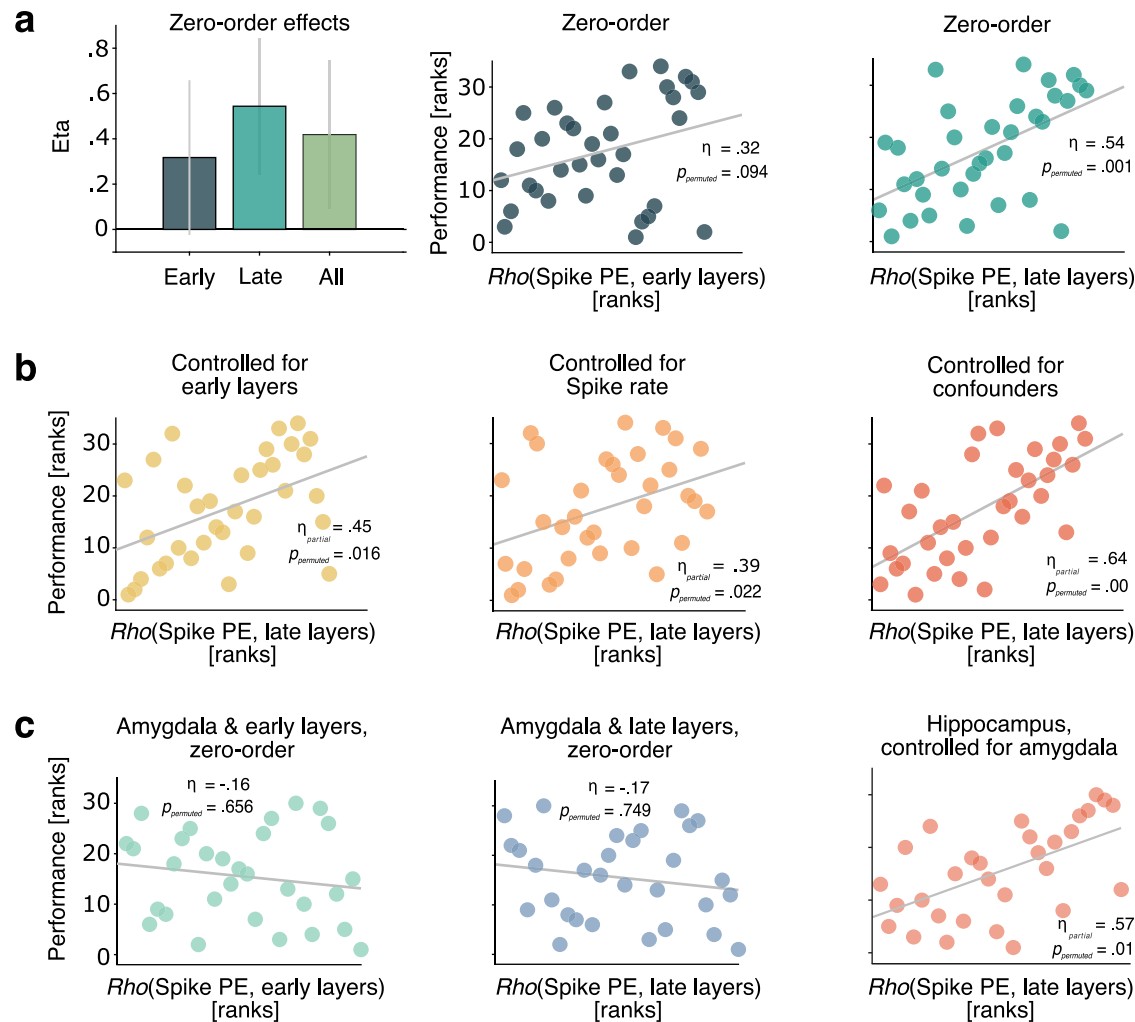

**Fig. 3 | Coupling of hippocampus spike entropy (PE) to late-layer features specifically predicts memory performance.** All *p* values have been computed using non-parametric permutation tests (see Online Methods). **a**: Zero-order eta estimates for early-layer, late-layer, and all-layer coupling vs. performance (left, bar graphs including bootstrapped 95% CI). Recognition performance was captured by a principal component score that combined accuracy, dprime and confidence. Note that criterion was not related to spike PE coupling (Fig. S3), represents bias rather than performance, is very weakly correlated with all performance measures (see Methods), and was hence omitted in this case. Zero-order relationship between hippocampal spike coupling estimates (individual latent correlations) and recognition performance (principal component capturing accuracy, dprime, and confidence) for early-layer models (middle) and late-layer models (right). All correlations are non-parametric (Spearman; *n* = 34). **b**: Unique links of hippocampal spike coupling to late-layer features after controlling for coupling to early-layer features (left), coupling of spike rate to late-layer features (middle), and a set of other potential between-subject confounds (number of neurons and trials, task variant, encoding duration, age; right). **c**: Effects are specific to the hippocampus. Coupling of amygdala spike entropy to either early-layer (left) nor late-layer features (middle) did not predict recognition performance (*n* = 30). Finally, controlling for amygdala spike PE coupling to late-layer features did not reduce the link of hippocampus late-layer coupling to performance (right).

aggregated (late-layer) feature content of each specific stimulus at encoding, better memory performance becomes feasible. Variability (PE) may thus provide a core signature of item-specific memories in hippocampus.

To embed our findings in the context of broader theories of memory encoding, it's essential to acknowledge the central role of the PFC in nearly every classic memory encoding systems account, permitting functions as broad as cognitive control[30–32] and working memory[33,34] during memory formation. An interesting alternate account of PFC function during encoding suggests that while the hippocampus may capture dynamic changes in the spatial and temporal aspects of incoming inputs, the (medial) prefrontal cortex may sort those inputs based on their similarity and integrate them over time[35]. This function would allow the mPFC to determine the most relevant (and valuable[36]) content of our encoded experiences[35], an indispensable aspect of effective learning and decision-making. From

this perspective, one could predict that PFC would provide sort-and-integrate functionality most efficiently by using aggregated, later-layer stimulus features rather than from simpler features, just as we propose for hippocampus. For example, one could investigate whether the PFC may sort inputs by testing whether images with similar late-layer feature content also express similar levels of neural variability. Future work could test these ideas using recordings from both hippocampus and PFC in the context of neural variability during memory encoding.

Crucially, that trial-level mapping between hippocampal spiking variability and visual content predicted memory formation success over and above standard spike rate[37] in our data further buttresses a growing literature revealing the unique behavioural relevance of moment-to-moment fluctuations in brain activity[14]. Indeed, we and others have argued that control processes may flexibly adapt neural variability (so-called "meta-variability") to meet the resource demands of a given task, thereby enabling optimal behaviour[14,18,38]; here, we

show that visual feature-driven meta-variability is required for memory success. Although promising, the very idea of meta-variability requires new theories and tools that elucidate the behavioural relevance of within-trial temporal neural variability beyond typically used measures in neuroscience (e.g., the across-trial Fano factor)[14].

Finally, using our freely open and available methodological framework (see Methods), future research could test alternative models of conjunctive representations in hippocampus in a within-participant, across-trial manner. For example, one could test the presence of object category representations[39], or of any map-like representation spanning space[40], direction[41], or non-spatial relational maps[42] within and beyond the hippocampus or MTL (e.g., prefrontal cortex). Importantly, our approach permits the estimation of any joint space between neural activity on the one side and multivariate stimulus properties of any kind on the other, for each subject. Doing so allows the optimal expression of individual response profiles that can subsequently be compared across subjects in any desired context, regardless of recording specifics (e.g., exact cells, locations). Crucially, by decomposing this shared space between neural activity and stimulus features, one estimates a low dimensional representation of how neural responses represent stimulus properties of interest, an approach that is immediately complementary to recent large-scale efforts to summarize neural activity alone using dimensionality reduction techniques[43,44].

Overall, we propose that moment-to-moment spiking variability provides a new window into how the hippocampus constructs memories from the building blocks of our visual world.

## Methods

### Sample and electrophysiology
We re-analysed human hippocampus and amygdala single-neuron activity from a previously published dataset[20] of 42 patients (16–70 years; 15 female; total number of sessions = 65) undergoing surgery for intractable epilepsy who performed an encoding and recognition memory task (see below). Electrodes were localized based on post-operative MRI images and locations were only chosen according to clinical criteria. Protocols were approved by the institutional review boards of the Cedars-Sinai Medical Center, Huntington Memorial Hospital, and the California Institute of Technology. Informed consent was obtained from all participants. We analysed the same single units that were isolated using spike sorting for an earlier release of this data set[20], and we focused on spikes fired within the first 1000 ms of stimulus presentation during the encoding phase of the task.

### Task
Patients were first presented with images from 5 out of 10 possible categories (across task variants) during an encoding phase (1–2 sec; 100 trials) and performed an animacy judgment (animal vs. not; unlimited time to respond) on these images. After a 15–30 min delay, they were presented with a set of images that contained both previously seen and novel images (50% each; 100 trials) and were asked to simultaneously judge images as old or new and provide a confidence rating (from 1- new/confident to 6- old/confident). While further details on task, recordings, and basic performance can be found elsewhere[20,45], it is important to note that we limited our analyses of neural activity to the first encoding session of $n = 34$ patients whose recordings included active hippocampus neurons (average neuron PE > 0.0001; 12 ± 11 hippocampal neurons per individual and session, total $N = 411$). The analysis of neural activity in the amygdala included $n = 30$ patients who also had active neurons in the amygdala. Memory performance was quantified using across-trial behavioural data from the corresponding recognition session, for which we focused on recognition accuracy, dprime, confidence, and confidence-weighted accuracy, while additionally including response criterion. We analysed absolute confidence by collapsing across old and new decisions, resulting in confidence values of 1–3 (low to high confidence) that were

averaged within participants and across trials. Of note, all primary results are based on a principal component score of performance which was generated via a PCA on all metrics but response criterion (eigenvalue = 2.8, [standardized loadings = 0.89, 0.92, 0.92, 0.53] for accuracy, confidence-weighted accuracy, dprime, and confidence). Note that we did not include criterion in the PCA estimation because it represents response bias rather than performance and is weakly correlated with all other performance measures (avg$_{corr}$ = 0.064, ranging from −0.02 to −0.13). Separate analyses for each performance metric can also be found in the supplemental material.

### Using computational vision models to estimate the content of stimuli participants were asked to encode
With the goal of estimating image features at different levels of aggregation, from simple, orientation-like features to more complex, composite features, we employed two different computational vision models, HMAX[13] and VGG16[22]. Both models are openly available and have previously been used to estimate image content at different aggregation levels[18,27].

### HMAX
The HMAX model is a biologically-inspired, feedforward model of the ventral visual stream that contains four hierarchical layers, S1, C1, S2, and C2[13]. S1 and C1 layers correspond to visuo-cortical areas V1/V2, whereas S2 and C2 correspond to V2/V4[46]. Within the first layer (S1) each unit is modelled with a different Gabor filter. These filters vary with respect to their orientation (HMAX defaults: −45°, 0°, 45°, 90°) and their size, the n × n pixel neighbourhood over which the filter is applied (sizes: [7:2:37]). The resulting activation map of the S1 layer contains the simple cell responses for every position within the input image. Next, each C1 unit receives the result of a maximization across a pool of simple S1 units with the same preferred orientation but with (a) varying filter sizes and (b) at different positions (spatial pooling). We used 16 filter sizes at the first layer and maximized only across adjacent filter sizes, resulting in 8 scale bands. In the following, S2 units merge inputs across C1 units within the same neighbourhood and scale band, but across all four orientations. Importantly the response of S2 units is calculated as the fit between input and a stored prototype. At the final layer (C2), a global maximum across positions and scales for each prototype is taken, fitting eight C1 neighbourhoods [2:2:16] using 400 different prototype features[46]. For all images seen by participants during encoding, we extracted estimates for C1 and C2 layers for further analysis of within-subject coupling between image features and spiking variability (see below).

### VGG16
Additionally, we processed images using VGG16, one of the most commonly used convolutional neural networks of computer vision, characterized by its high number of convolutional layers and its very high accuracy in object classification[22]. Here, each input image is processed by a stack of 13 convolutional layers, with stride and spatial padding of one pixel and a receptive field of 3 × 3 pixels. The number of features per convolutional layer gradually increases from early to late layers ([64, 128, 256, 512, 512]). Convolutional layers are interleaved with five max-pooling layers that carry out spatial pooling[22]. The stack of convolutional layers is followed by three fully connected layers and one soft-max layer. We used VGG16 as implemented in TensorFlow, pre-trained on the image-net dataset[47,48]. For each image that participants encoded during the experiment, we extracted predicted feature (heat) maps for max-pooling layers 1–5 (corresponding to layers 3, 6, 10, 14, and 18) for further analysis of within-subject coupling between image features and spiking variability (see below).

### Extraction of image features from layer-wise feature maps
We estimated image features by extracting three feature summary metrics from each model layer and image: (1) the spatial sum and (2)

spatial standard deviation (SD) from C1 and C2 layers (HMAX) and from max-pooling layers 1–5 (VGG16), as well as (3) the number of zero elements (i.e. pixels) for VGG16 max-pooling layers 1–5 (note that this latter feature metric was not estimable from HMAX due to a near complete lack of zero elements from its layer-wise model output for images used in our stimulus set). These three feature metrics were chosen to arrive at a comprehensive approximation of image features that incorporates overall saliency (spatial sum), the distribution of salient and non-salient image locations (spatial SD), and the sparsity of saliency maps (number of non-zero entries). Additionally, note that the spatial sum and SD were only computed across non-zero map elements.

Importantly, the number of features varies across the layers of HMAX (C1 = 32, C2 = 400) and VGG16 (layers 1–5: 64, 128, 256, 512, and 512, respectively). To ensure maximal comparability across layers, we did the following, separately, for each model layer and image statistic (spatial sum, standard deviation, number of non-zero entries): We entered the metrics from all feature maps and all images to PCA and extracted the first principal component score (this, in effect, puts all images on the same scale for further analysis). Thus, each image participants saw at encoding was represented by three principal component scores for each model layer of interest, one each capturing the layer-wise spatial sum, standard deviation, and number of non-zero entries. These scores subsequently served as input for individual PLS models to estimate the coupling between image features and spiking variability (see below).

### Estimation of spiking variability (permutation entropy)

For each single unit and trial during the encoding phase of the memory task, we extracted the first 1000 ms after stimulus onset in non-overlapping bins of 10 ms length and extracted the bin-wise spike counts. Based on the resulting spike trains, we then calculated permutation entropy (PE) for each neuron and trial[49] to measure the temporal variability of neuronal responses during encoding. Note that permutation entropy is tailored for analyses of this kind as it does not come with distributional assumptions and has been designed with physiological data in mind. We applied PE instead of more commonly used estimates of time series variability (e.g., standard deviation, Fano factor) due to the special distributional properties of single-trial spiking data that often violate normality assumptions (due e.g., to extreme sparsity).

To calculate permutation entropy, a time series is first partitioned into overlapping sections of length $m$[21]. The data in each section is then transformed into ordinal rankings so that every section is represented by a unique pattern. For example, the sequence (2,11,14) corresponds to the pattern (0,1,2), whereas the sequence (15,19,1) maps to (1,2,0). Thereafter we can count the relative frequency $p_i$ of all patterns and compute PE as:

$$PE_m = - \sum_{i=1}^{m!} p_i \log_2 p_i \qquad (1)$$

where m corresponds to the length of sections and $m!$ describes the number of possible patterns. We computed PE for three different motif lengths ([2,3,4]). Neuron- and trial-wise PE estimates of all three motif lengths were used within individual partial least squares (PLS) models to estimate the individual coupling of image features to spike entropy (see below).

We used the Matlab implementation of EntropyHub 2.0[49] to calculate the permutation entropy.

### Estimating the within-person coupling of image features and spiking entropy

To quantify the individual multivariate relation between spiking entropy and image features of the presented images, we employed a behavioural partial least squares (PLS) analysis for each subject[19,50].

Here, PLS first calculated the rank correlation matrix (Rho) between the trial-wise estimates of stimulus features (e.g., C2$_{sum}$, C2$_{SD}$,

VGG$_{sum}$, VGG$_{SD}$, VGG$_{nz}$ for layers 3–5) and the trial-wise PE estimates of each recorded neuron, within-person (Fig. 1a). All neurons included had PE > 0.0001 and all trials included contained at least a third of neurons spiking at least once. Thresholding on the activity is necessary due to the notoriously sparse coding scheme in the hippocampus which including many trials and neurons which show no activity eventually conceals relevant relations between neural activity and sensory stimuli. This is a computational issue arising when the input matrix of the PLS contains predominantly zeros. Thus, for the latent coupling estimation to be meaningful we must ensure a base level of activity across trials and neurons. Above that baseline, different cut-offs left results qualitatively unchanged.

The Rho matrix was subsequently decomposed using singular value decomposition (SVD), generating a matrix of left singular vectors of image feature weights (U), a matrix of right singular vectors of neuron weights (V), and a diagonal matrix of singular values (S).

$$SVD_{Rho} = USV' \qquad (2)$$

The application of these weights yields orthogonal latent variables (LVs) which embody the maximal relation between feature content of the input and neural spiking entropy. The latent correlation of each LV is calculated by first applying neural weights to neuron-wise PE data and stimulus feature weights to the matrix of stimulus feature metrics, respectively, before correlating the resulting latent scores (Fig. 1d). Bootstrapping with replacement was used to estimate confidence intervals of observed latent correlations (1000 bootstraps). Importantly, given the variable and small number of trials and neurons across individuals, non-parametric Spearman correlations were used within PLS and throughout all other analyses.

To test the differential coupling of spike PE to various image features at different levels of image feature aggregation, we obtained individual coupling estimates for early layers of computational vision models (C1$_{sum}$ and C1$_{SD}$ from HMAX; VGG$_{sum}$, VGG$_{SD}$, and VGG$_{nz}$ for layers 1–3), late-layers (C2$_{sum}$ and C2$_{SD}$ from HMAX; VGG$_{sum}$, VGG$_{SD}$, and VGG$_{nz}$ for layers 3–5), and all layers (C1&C2$_{sum}$ and C1&C2$_{SD}$ from HMAX; VGG$_{sum}$, VGG$_{SD}$, and VGG$_{nz}$ for layers 1–5). This split of layers was performed to keep the number of features within each latent model constant while at the same time separately estimating the coupling of spike variability to early layer, late layer, and all image features. To quantify whether spike PE captures unique, behaviourally relevant aspects image feature coupling compared to what may be captured by spike rate, we also ran the same PLS models, but replacing spike PE with trial-wise spike rates. Additionally, to explore the topological specificity of memory-relevant spike-feature coupling, we computed separate PLS models for neurons recorded in hippocampus and amygdala, respectively.

### Statistical analyses

We compared the individual strength of spike PE coupling to early and late layers via a Wilcoxon signed rank test on the absolute latent correlations derived from PLS models based on early and late layers, respectively.

We then used linear models to regress the performance score (see above) onto individual latent estimates of coupling between spike entropy and image feature metrics (all in rank space). First, we ran zero-order models based on the spike PE coupling estimates of early, late, and all-layer PLS models, respectively. Next, we tested the unique explanatory power of late-layer coupling by separately controlling for early and all layer coupling estimates (and vice versa; see Fig. 3). To additionally contrast the behavioural relevance of spike PE coupling with more established metrics of single cell activity, we controlled effects of late-layer spike PE coupling for late-layer spike rate coupling. Finally, we controlled effects of late-layer hippocampal spike PE coupling for a set of inter-individual control variables (number of trials,

number of neurons used within analysis, task variant, encoding duration, age). For each model we computed estimates of partial eta, marking the unique portion of variance in performance explained by the relation modulation of hippocampal spike PE.

For all analyses, permuted *p* values were computed as the proportion of permutations revealing a higher eta estimate than our estimate from the original, unpermuted data (see Fig. 3 for all primary results). To this end, we randomly permuted the trial-wise PE values at encoding (1000×) for each patient and then computed (1000×) the subject-wise encoding-based coupling between spike PE and image features derived from the vision model layers (early-layers, late-layers, all layers) using PLS. The resulting coupling values were then used to rerun (1000×) the between-subject regression of recognition performance on the encoding-based coupling of spike PE and image features, yielding a null distribution of eta values. In the same manner, we computed permuted *p* values of the control analysis of the coupling between spike PE and late-layer image features, controlling for the set of inter-individual control variables (number of trials, number of neurons used within analysis, task variant, encoding duration, age). For the subsequent analyses controlling for the coupling of spike PE to early-layers and the coupling of spike rate to late-layers, we also ran 1000 permutations of these controls using the same permutation orders as for the spike PE to late-layer model to ensure direct comparability. Then we used the resulting coupling estimates of these covariates to control the linear model of recognition performance and late-layer coupling, yielding a distribution of 1000 partial etas from which a permuted *p* value was obtained. Finally, to probe the topological specificity of our findings, we further modeled performance as a function of amygdala spike PE coupling for late-layers and additionally controlled the effects of late-layer hippocampus coupling for amygdala effects, computing permuted *p* values as described above.

All statistical analyses, PE and HMAX estimation were run in MATLAB 2020a and VGG16 was run in python 3.0.

### Reporting summary
Further information on research design is available in the Nature Portfolio Reporting Summary linked to this article.

## Data availability
Analysed data have been published previously and can be downloaded (https://europepmc.org/article/pmc/pmc5810422).

## Code availability
Code to reproduce all main results is available at: https://doi.org/10.5281/zenodo.13827812.

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

## Acknowledgements

We thank Morgan Barense, Ulrich Mayr, and Markus Werkle-Bergner for fruitful discussions on earlier versions of this work.

L.W., F.K. and D.D.G were partially funded by an Emmy Noether Programme grant from the German Research Foundation (to DDG) and by the Max Planck UCL Centre for Computational Psychiatry and Ageing Research, Berlin. FK was partially funded by the Max Planck School of Cognition, Leipzig. U.R. was partially supported by NIH NINDS U01NS117839.

## Author contributions

In line with the CRediT framework[51], author contributions are listed as follows: Conceptualization was done by L.W. and D.D.G. The Methodology was developed by L.W., F.K., S.K. and D.D.G. Software was written by L.W., F.K., S.K. and D.D.G. Validation was carried out by L.W., F.K. and D.D.G. Formal analysis was conducted by L.W., F.K. and D.D.G. The investigation was performed by L.W., F.K., U.R., and D.D.G. Resources were supplied by L.W., F.K., U.L. and D.D.G. Data curation was managed by L.W., F.K. and U.R. Original draft was written by L.W., F.K. and D.D.G. Writing, review, and editing were completed by L.W., F.K., E.v.d.E., U.L., U.R. and D.D.G. Visualization was done by L.W. and F.K. Supervision was done by U.R. and D.D.G. Project administration was handled by L.W. and D.D.G. Funding acquisition was secured by U.L., U.R. and D.D.G.

## Funding

## Competing interests

The authors declare no competing interests.
