## [Transparent Peer Review file · Nature Communications]

Single-neuron spiking variability in hippocampus dynamically tracks sensory content during memory formation in humans

Corresponding Author: Dr Douglas Garrett

Version 0:

Reviewer comments:

Reviewer #1

(Remarks to the Author)

In this paper the authors report that there is a relationship between the level of coupling between the entropy in neural firing in the human hippocampus and high level image stimuli and the subsequent recall of the stimuli. This is a potentially interesting finding; however, this reviewer cannot recommend publication due to several concerns.

The first is that it is unclear whether the analysis pipeline used here will obtain spurious results due to pre-selection of the data. The primary analysis here is a very abstracted version of the data and can easily be subject to this type of issue. On page 9, the methods state that "All neurons included had PE > .0001 and all trials included contained at least a third of neurons spiking at least once (different cut-offs left results qualitatively unchanged)," and later that the bootstrapping was performed on these. This raises a critical methodological question -- is it possible that this limitation to neurons with a high PE and limitation of trials is selecting precisely that set of data which will end up being significant in the subsequent testing? It is unclear from this description whether this represents a form of pre-selection or enriching of the dataset which lead to spurious apparently significant results in subsequent testing (see Kriegeskorte et al., 2009, Nat Neurosci, 12, 535-40; Thorp and Steinmetz, 2007, Computational and Systems Neuroscience). This could be clarified with a clearer description of the methods and perhaps a sort of flow diagram of how the data is processed and selected. Assuming there has been pre-selection, this can be corrected with appropriate uses of data subsetting or simply running the analysis without that pre-selection.

A second major concern is that the authors miss an opportunity to tie these results into broader theories of how human memory works and which brain areas are involved. The differences described between the hippocampus and amygdala here seem to compare nicely with those obtained using spike rate in Urgolites et al., 2022, Proc Natl Acad Sci U S A, 119. They also miss the opportunity to relate these results to the very large MRI literature. The latter seems important in this study given the large number of "neurons" which were isolated. How does averaged neural activity compare to the results obtained with a BOLD signal.

Finally, the major point which is being made appears to be obscured by very vague language and formulation. For example on page 2, what does the sentence "The processing of different visual features indeed uniquely impacts..." mean? It seems vague.

In summary, it seems like there are potentially some interesting findings here, but this paper would benefit from a more careful consideration of existing theories of human memory and a more thorough explanation of exactly how the analyses are being performed.

Following are some additional more detailed points which the authors may wish to consider in revision.

How do the present results relate to the previously described dependence of neural responses in these areas to low level image properties like brightness and contrast (Steinmetz et al., 2011, Journal of Neurophysiology, 105, 2874-2884). What happens to the relationship to entropy coupling if just the corresponding elements of the current models are considered?

p1 para 2 "conjuncted" - should this be "conjoined" or even "joined"

Unclear what is meant by the sentence "Beyond the behavioral relevance..."

"We posit that..." This sentence also seems vague because "stronger trial by trial coupling" is ill defined.

page 3 - 12 +- 11 hippocampal neurons per subject seems high for well isolated units, though this clearly depends on number of recording channels. Most investigators report perhaps 0.3 - 0.5 well-isolated neurons per channel of recording. It would be good to report average numbers per recording channel and numbers of recording channels here.

p5 "predicted memory formation success over and above the standard spike rate" - how was this demonstrated in any way? It does not appear there was any direct comparison to this alternative model. This is a critical question - does the measure of entropy provide any further information regarding memory formation than firing rate measures?

Reviewer #2

(Remarks to the Author)

The authors re-analyze an existing dataset of spiking activity recorded in the human medial temporal lobes of epilepsy patients that perform a long-term memory task. In this task, patients are viewing images of objects for encoding. Fifteen to 30 minutes later, the patients are presented with old and new images and are asked to judge whether they saw the images earlier or not, and how confident they are in their memory judgment. Spiking activity is analyzed by computing the trial-wise permutational entropy (PE) of spike trains during encoding. PE is then correlated with stimulus features extracted from early and late layers of artificial neural networks trained to classify images of objects (spatial Sum, SD, and sparsity of the activation patterns in a layer). Correlations of spike PE with early- and late-layer features of the images across trials are computed for each individual patient. These correlation scores are then correlated with a behavioral memory performance score. Late- but not early-layers correlation with PE correlate with behavioral performance. The authors conclude that participants with high coupling of spike-train-variability (spike PE) with late-, and early-layer features profit from more differentiated neuronal activity between stimuli that benefits memory encoding and suggest spike PE as novel measure to analyze neuronal activity.

Overall, I think that the study reads very well and provides an important addition to the literature on the neuronal basis of long-term memory by introducing trial-to-trial spike-train variability in the hippocampus as novel measure that explains inter-individual variance in memory performance.

I have two main points that I think should be addressed. The first one relates to the interpretation of the findings. The authors conclude in the first paragraph of the discussion that "[...] findings of more limited (but still present) behavioral relevance of early-layer features suggest a hierarchy in which the hippocampus downweights simple features without discarding them entirely". The overall pattern of results, however, seem to me to disagree with the conclusion that spike PE coupling to early-layer features significantly correlates with behavior.

The only result that could support such a conclusion is that individual estimates of coupling of Spike PE with early layer features alone are larger than zero (e.g., Fig. 2b, yellow dots), if I understand correctly. Here, the authors correlate spike PE with early and late features in neural nets on an individual subject level. They find, that both early and late features significantly correlate with spike PE and that correlations of late features with PE are higher than correlations of early features with PE. Furthermore, these correlation scores on the subject level (early features with PE, late features with PE) are correlated with one another across subjects. One question I have here is whether it could be that features of early and late layers are correlated within the neural nets as well. If so, this would probably need to be partialled out. In any case, these correlations do not include behavior and hence do not directly support the conclusion above.

The correlations of spike PE coupling to early-layer image features with behavior are all statistically insignificant. The authors nevertheless write "Later memory performance was positively correlated with the coupling of hippocampal spike variability during encoding to image features for early-layers (Fig 3a, $h = .32$, $t_{31} = 1.9$, $p = .07$) [...]" (see also Fig. 3b). If one follows a frequentist approach and assumes $p < .05$ as threshold for significance, there is no significant effect in this case, if I understand correctly. In Supplementary Figure 3, it seems also that measures relating to memory performance (accuracy, d_{prime}) are only positively correlated with late-layer to spike PE coupling.

Most importantly, the in my view most informative analyses in that respect speak against a meaningful relationship of spike PE coupling to early-layer features to behavior. These are the analyses that control for influences of late-layer features that render the association of coupling early-layer features to spike PE with behavior insignificant. Correlations of coupling of late-layer features to spike PE with behavior, in contrast, survive after controlling for coupling of early-layer features to spike PE (Page 5, first paragraph).

Taken together, it seems that the pattern of results fits better with the conclusion that significant coupling of late-layer features to spike PE in the hippocampus is found to correlate with behavior, while correlations of coupling of early-layer features to spike PE with behavior do not survive more conservative analyses controlling for other influences, most importantly the influence of coupling of spike PE with late-layer features. This pattern of results hence supports notions that *solely* representations of more composite stimulus features in hippocampus are behaviorally relevant in my reading. I think this conclusion goes well with the notion that hippocampus operates using an invariant, semantic code as most famously exemplified by the classic findings on concept cells (Quiari Quiroga et al. 2004, Nature), and findings where their invariance also extends from the visual in the written and auditory domain as shown in e.g., Quiari Quiroga 2009, Current Biology). While these studies do not compare semantic (late layer) vs. more perceptual features (early layers), I contributed to a study

where representational similarity of population activity was explained by abstract, categorical features of visual stimuli while low-level visual features of the stimuli could not explain patterns in neural activity (Figure 2 and Supplemental Figure 2 in Reber et al. 2019, PLoS Biology).

The other major point relates to comparing intra- vs. interindividual approaches to analyze their data. Page 1, bottom, the authors write “We posit that individuals with stronger trial-by-trial coupling between hippocampal variability and the content of visual input should also exhibit superior memory, providing evidence that visual features have successfully been encoded by the hippocampus during memory formation.” Focusing on interindividual variance in spike PE across participants to explain interindividual variance in performance is perfectly fine and interesting in my opinion. To get a more complete picture, however, I suggest conducting subsequent memory analyses and compare coupling of spiking PE with features of neural nets during presentation of later remembered vs. forgotten items, as well as those remembered with high vs. low confidence within participant. This would give us an idea of whether the reported phenomenon is indeed something trait like and/or whether it fluctuates also within participant to explain intraindividual differences in performance.

Minor points

The authors use a principal component score of various behavioral measures such as accuracy, d-prime and confidence. These measures, however, measure distinct concepts and I do not see the advantage of a composite score. This information is provided in given in Figure S3. There, the reader can appreciate that both accuracy and dprime correlate with late layers but not with early layers, while it seems to be more the case for confidence to be significantly correlated with early- and late-layers couplings to spike PE. I think this should go in the main text and figures.

In Figure 3B and Figure 3C, rightmost panels look like the data going in the plots are identical, but the stats reported on the figure differ.

Reviewer #3

(Remarks to the Author)

Waschke et al. claims that the more precisely hippocampal spiking variability tracks the composite features that comprise each individual stimulus, the better those stimuli are later remembered. The topic of this research is significant to the field of neuroscience as it aims to develop a linking model between sensory encoding and memory formation.

However, to support this claim the authors need to do the following:

1. Reasonably operationalize what they mean by composite features of individual stimulus
2. Reasonably quantify spike variability and operationalize what they mean by “tracking” the entity defined in 1.
3. Reasonably quantify “remembrance” of stimuli and link that to 2.

Overall, while the authors have provided attempts to address all 3 of these aspects, I have struggled to fully check off these points while reading the paper. Therefore, while the overall topic is very interesting, the evaluation of this manuscript, from my end, remains incomplete due to a significant lack of clarity to motivate and demonstrate the evidence supporting the claims. A major revision should be able to make the manuscript clearer and allow for better re-evaluation.

Major comments:

1. Are the authors claiming that hippocampal spike variability is partially explained by image stimulus features (in other words ventral stream activity drives the spiking variability in the hippocampus)? And the better it is (at a subject level), the higher performing the subjects are in a memory task? If so, I didn't find any evidence of eye tracking done in the study during the initial encoding phase. For instance, if some subjects simply didn't look at the stimuli (time to time), they will demonstrate a lower link between image features and hippocampal activity (whichever way it is summarized) and a lower performance in the memory tasks. In the absence of eye tracking data, it is important to show that the performance of the participants in the animacy judgement task (during encoding) do not vary significantly and do not correlate with the memory task performance (even though the confound still remains to some degree).
2. Related to the above point, one critical information about the memory task performance is the reliability of the behavioral data per subject. Was there an independent measure of performance reliability? A subject with unreliable data, typically demonstrate lower performance and lower correlation with any entity that is trying to predict that performance.
3. The paper is somewhat poorly written, in the sense that the key concepts (especially with relation to the two randomly chosen models HMAX and VGG, and the quantification of feature statistics) are not at all explained and motivated clearly. As a reader I struggled to understand how they relate undefined terms like “composite features that comprise each individual stimulus” to network activation maps, and the first PC etc. It remains so cryptic that a genuine review of that claim isn't feasible. It is not hard to establish that semantically meaningful constructs can be better linearly separated from features of later layers of CNNs (compared to its earlier layers). If this is the phenomena in the model that the authors are looking to couple with the hippocampal neural spike variability, then there might be more direct ways of establishing it.
4. Establishing whether hippocampal spike variability is better driven by early vs. late layer features of the models can also be done and demonstrated with more standard RSA or linear prediction type methods. Correct neural ceilings need to be estimated and investigated for these types of comparisons to accurately depict how strong these coupling strengths are.

More detailed information of how choices in the number of components used from the PCA analysis per layer etc. affect the results is critical.

5. As far as I can tell, no alternate hypothesis of hippocampal activity statistics have been tested. For example, if the analysis is repeated with overall activity levels per trial instead of PE per trial, do we get similar results? I think some analysis of this sort is critical to claim that spike variability is the only key mechanism at play. The authors do test whether controlling for trial-level spike rate change the link between memory performance and the coupling between late layer features and spike variability (Fig 3B, mid panel), but they should also directly report the coupling between trial-level spike rate and late layer features (e.g. the y-axis of Fig 2B, but spike PE replaced by spike Rates).

Version 1:

Reviewer comments:

Reviewer #1

(Remarks to the Author)

The authors have expanded the descriptions of their methods so that the analysis pipeline is clearer. Now that this is clearer, I believe that unfortunately this analysis as well as the additional analyses performed show that the results are not statistically significant. This is because when the fraction of neurons having a PE < 0.0001 is increased, the authors report they do not have enough remaining data to achieve a significant result.

While the authors consider the impact of this pre-exclusion and the potential rationale for doing so, it appears the analysis does not consider the fundamental point of both Kriegeskorte et al 2009 and Thorp and Steinmetz 2007. That is - when you pre-select a set of data based on one statistical test and then use that subset in a subsequent test, you often have biased the distribution of that data so that your estimate of the significance level, or likelihood of having a type-I error, in the subsequent test is grossly incorrect, potentially by up to 1800%.

In this case, the authors select neurons based on the PE and then subsequently used a statistic derived from the PE, namely computing the PLS between the PE and image features. The relationship between the PLS and subsequent memory performance is one of the primary claims of this paper.

There are several ways in which this issue can normally be addressed. One is to choose neurons based on the PE for one randomly selected subset of trials and then compute the derived statistic based on the other trials. The report of a similar analysis in the response to reviews suggests this will lose so much power that a significant result will not be obtained, though the authors do report such a test directly.

The other approach is to apply either the same or a different statistic which perhaps represents more directly what the authors wish to measure to all neurons on all trials and see if a significant result is obtained. Again the description of the subsetting reported in the response strongly suggests that this will not produce a significant result.

The results are certainly interesting and this sort of pre-selection is actually rather common in neuroscience. However, without actual demonstration of a significant result, this reviewer cannot recommend publication.

Reviewer #2

(Remarks to the Author)

I thank the authors for carefully replying to the comments made. All points I raised are sufficiently addressed in my view.

Reviewer #3

(Remarks to the Author)

I thank the authors for significantly improving the manuscript by addressing each of my previous comments in great detail in their revised version. As such, I have no further comments on the article and would recommend it for publication.

Version 2:

Reviewer comments:

Reviewer #1

(Remarks to the Author)

I have reviewed the second revision of the paper entitled "Single-neuron spiking variability in hippocampus dynamically tracks sensory content during memory formation in humans".

The authors have further clarified the independent nature of the observations of relationship between spiking variability and image features and subsequent memory performance. Given that the random variation in these is independent, my primary concerns have been addressed.

The authors have also added additional permutation tests which strengthen their results regarding the relationship between spiking variability and image features.

REVIEWER #1 (Remarks to the Author):

Remark 1

In this paper the authors report that there is a relationship between the level of coupling between the entropy in neural firing in the human hippocampus and high-level image stimuli and the subsequent recall of the stimuli. This is a potentially interesting finding; however, this reviewer cannot recommend publication due to several concerns.

The first is that it is unclear whether the analysis pipeline used here will obtain spurious results due to pre-selection of the data. The primary analysis here is a very abstracted version of the data and can easily be subject to this type of issue. On page 9, the methods state that “All neurons included had PE > .0001 and all trials included contained at least a third of neurons spiking at least once (different cut-offs left results qualitatively unchanged).” And later that the bootstrapping was performed on these. This raises a critical methodological question – is it possible that this limitation to neurons with a high PE and limitation of trials is selecting precisely that set of data which will end up being significant in the subsequent testing? It is unclear from this description whether this represents a form of pre-selection or enriching of the dataset which lead to spurious apparently significant results in subsequent testing (see Kriegeskorte et al., 2009, *Nat Neurosci*, 12, 535-40; Thorp and Steinmetz, 2007, *Computational and Systems Neuroscience*). This could be clarified with a clearer description of the methods and perhaps a sort of flow diagram of how the data is processed and selected. Assuming there has been pre-selection, this can be corrected with appropriate uses of data subsetting or simply running the analysis without that pre-selection.

- **RESPONSE:** Thank you for raising this important point. We agree that ‘double-dipping’ (Kriegeskorte et al., 2009, *Nat Neurosci*) is an important concern in research in general; circularity indeed arises if the selection criteria and final results are not independent. In the single-neuron domain here, this would translate into preselecting only those *within*-subject neurons which show outcome-specific response characteristics prior to estimating the association between neural activity and that same outcome of interest *across* subjects (Kriegeskorte, Simmons, Bellgowan, & Baker, 2009). Crucially, this is not a concern in the present study, as we elaborate in detail below.
- To fully communicate our motivation for the choices for data handling in the current study, we begin by better describing our modelling approach. Our research question centers on the effect of trial-to-trial differences in sensory input on spiking activity in the hippocampus; therefore, instead of aggregating over trials to identify neurons carrying a general memory signal (as is done in many previous spiking studies (Rutishauser et al., 2015; Urgolites et al., 2022; Wixted et al., 2014)), we use partial least squares correlation (PLS) to estimate the maximal trial-wise association between neural activity and sensory input during memory encoding (which in our data later predicts recognition memory performance) by projecting both data matrices into a joint latent space (Krishnan, Williams, McIntosh, & Abdi, 2011).
- The basis for our within-subject PLS (see Fig R3 for a detailed description) is a decomposition (SVD) of the rank correlation matrix between each neuron and each image feature, and each correlation within this matrix is computed *across trials*. For PLS to be viable, we need to ensure that these correlations are estimated using at least a reasonable proportion of non-zero trial data. Because “image features” exist for every trial (i.e., each image at encoding is run through the HMAX and VGG models, always yielding estimated image features), the only potential risk of “0” here is the Permutation Entropy (PE) level on each trial for each neuron.
- We approach detecting “PE = 0” in two complementary ways:
 - a) In our original submission, we had set an overall activity cut-off for each neuron as an “average, across-trial PE > .0001” at the initial stage of our pipeline, *prior* to the any statistical analysis of the data. However, we now realize that this can be written even more simply; we only keep cells that have an average, across-trial PE that is non-zero (i.e., PE > 0). Our intention here was to remove neurons from further analysis that exhibit no estimable PE at all, which is perhaps the most conservative approach that can be taken and corresponds to excluding neurons that do not display any spikes in the time-window of interest. Any standard spiking paper would typically be even more harsh, pre-determining statistically “active” cells (those whose spike rate is *statistically* differentiated from zero) prior to estimating primary statistical effects of interest (Faraut et al., 2018). Our approach

relies only on the numerical level of PE ($PE > 0$; we do not statistically test this quantity against zero or against another time window). This ensures that even very sparsely firing cells can be examined in relation to PE (as long as $PE > 0$). Such an activity minimum is also inherently agnostic/blind to the experimental design, the particular stimuli presented, or the behavioural outcome; as a result, double dipping is of no concern here.

- b) Next, after removing neurons with $PE = 0$, we set a criterion for a minimum number of “active” neurons *per trial* at 1/3rd, again *before any statistical analyses were done on the data*. This means that at least 1/3rd of all cells must have $PE > 0$ on that trial for that trial to be included in any further analyses. The motivation here is that because we compute latent trial-wise correlation between PE and image features, there needs to be at least a reasonable proportion of cells with non-zero PE for every trial (i.e., if no cells exhibit a $PE > 0$ on a given trial, latent level neural effects cannot be computed based on that trial by definition).
- Why is this important? Using results of a simulation analysis (see Fig R1), we depict the impact of greater numbers of $PE=0$ trial data on the estimation of the “true” underlying correlation between spiking PE and image metrics. Here, one can see that the more $PE=0$ values are present (e.g., from no threshold to 1/3 to 2/3), the greater the error within estimated latent correlations using PLS.
 - Thus, fully *removing* the within-trial threshold of number of neurons (thus allowing all trials to be included in the analysis for every subject) greatly increases the chances of within-subject model results being “contaminated” by zeros, increasing the error of any potential across-trial correlations between by the inclusion of null data.
 - Conversely, increasing the threshold to 2/3 yields the highest precision of the estimated latent correlations by deleting more zeros. However, this greater precision comes at extreme costs, as elaborated on below.

Figure R1: Simulation displaying the effect of trials with $PE=0$ on the estimated latent correlation. Correlations between one image feature and spiking PE of 12 neurons are simulated across 100 trials. The true underlying correlation is one with an added noise level of .8; in 15 trials, a random set of neurons is silenced. (a) Correlation of an example neuron and the image metric, showing the effect of silent trials in the native data space. (b) Estimated correlations in the latent (PLS) space; colours display the applied thresholds (green: no threshold, orange: 1/3 active neurons, blue: 2/3 active neurons). (c) Precision of estimated latent correlations (i.e., 1-latent correlation) across 500 simulation runs.

- As requested by the reviewer, we now closely investigate the overall impact of using *different* thresholds of the minimum within-trial neuron count on our overall results, a set of analyses we had not done prior to our original submission. We compare the impact of the original threshold (at least 1/3 of neurons must exhibit $PE > 0$) vs. fully removing (i.e., no minimum number of non-zero PE neurons within trial) or doubling (2/3 of neurons must exhibit non-zero PE) the within-trial threshold.

Trial count impact

- If we look at the individual level (Fig R2, left panel), one can see that increasing the threshold from 0 to 1/3 yields a dramatic reduction in the number of maintained $PE = 0$ trials in some subjects (e.g., subjects 1-4). Crucially however, every subject maintained enough trials to make across trial correlations between PE and image features viable (i.e., at least 30 per subject);

lower left panel). At our original threshold of 1/3, 94% of all trial data on average is maintained across subjects (Fig R2, lower left panel).

- Increasing the threshold further (from 1/3 to 2/3) reduces the numbers of possible PE = 0 responses on any trial for any neuron much more drastically (Fig R2, right panel). But as a result, a dramatic number of trials are lost using this threshold in our data, to a point that across-trial correlations would be estimated from far too few trials to be defensible (e.g., many subjects maintain less than 10 trials; lower right panel). The across-trial correlations for each neuron are supported by a median of 42/100 trials, a loss of 58% on average across subjects (Fig R2, lower right panel).

Figure R2: Effect of different within trial thresholds on the number of zeros removed from the data (relative to no threshold; upper panel) and the percentage of trials maintained for across-trial correlations between PE and image features, lines indicating the median trial count across subjects (lower panel).

Memory performance impact

- The impact of removing the within-trial threshold is also quite dramatic on memory effects in our data; no threshold (threshold = 0) yields a much smaller correlation with memory performance ($\eta = 0.28$), likely due to the far greater presence of null PE data contaminating the correlation with image features (see simulations above).
- The 2/3 criterion provides a correlation of $\eta = 0.47$ between latent coupling and recognition memory performance. However, due to the loss of 58% of trials we see no theoretical or evidence-based reason why a more extreme threshold of 2/3 is necessary, despite its greater ability to eliminate zeros from the correlation matrix; we argue that as much trial data should always be maintained as possible, wherever possible, limiting the chances of overfitting in any model scenario.
- Considering all evidence here, we thus argue that our previous choice of within-trial threshold (that 1/3 of neurons must have PE > 0 to keep that trial) in the manuscript maintains most trial data (median 94%), while minimizing zeros in the data matrix prior to estimating effects of interest. We argue that this combination of choices is perhaps the most reasonable choice that can be made in the current context.
- As suggested by the reviewer, we have revised several sections in the main text to make the steps of our analysis clearer and now also depicted the complete analysis pipeline as a flowchart in Fig R3. Although perhaps too detailed for the main paper, if the reviewer finds this new flowchart useful to understand our analysis pipeline, we would be happy to also add it to the supplemental material of the manuscript.

Figure R3: Flowchart of data handling and analysis steps

Remark 2

A second major concern is that the authors miss an opportunity to tie these results into broader theories of how human memory works and which brain areas are involved. The differences described between the hippocampus and amygdala here seem to compare nicely with those obtained using spike rate in Urgolites et al., 2022, Proc Natl Acad Sci U S A, 119. They also miss the opportunity to relate these results to the very large MRI literature. The latter seems important in this study given the large number of "neurons" which were isolated. How does averaged neural activity compare to the results obtained with a BOLD signal.

- **RESPONSE:** Thanks for these great points. We split our response into three parts: We start by discussing the study by Urgolites et al. (2022), then compare our findings to the fMRI literature and lastly link our findings to the more general memory theories
- Urgolites et al. (Urgolites et al., 2022) differentiated between two types of memory signals: A “generic” signal which was found in neurons of the hippocampus, amygdala, PFC and ACC, and a sparse, item-specific signal unique to neurons in the hippocampus. Here, we find a memory performance-relevant dissociation in how the hippocampus and amygdala respond to layer specific content in encoded images. It is feasible that our results provide a basis/mechanism for understanding *how* item specificity may exist in hippocampus; by modulating neural variability (PE) more tightly in line with the aggregated (late-layer) feature content of each specific stimulus at encoding, better memory performance becomes possible. Variability (PE) may thus provide a core signature of item-specific memories in hippocampus. We note this in Discussion, paragraph 4 as well:
 - Discussion (page 6, paragraph 4): *“The absence of memory-relevant spike variability coupling in the amygdala, despite this structure’s known role in memory formation (Rutishauser et al., 2011) and direct afferents from visual cortical areas (Price & Joseph, 2003), further highlights the unique role of the hippocampus during the generation of long-term memories. This dissociation between hippocampus and amygdala dynamics in our study is in line with recent findings showing that item-specific memory signals may only be present in the hippocampus, while more generic memory signals may also be present in the amygdala and other brain areas (Urgolites et al., 2022). It is feasible that our results provide a basis/mechanism for understanding how item specificity may exist in hippocampus; by modulating neural variability (PE)*

more tightly in line with the aggregated (late-layer) feature content of each specific stimulus at encoding, better memory performance becomes feasible. Variability (PE) may thus provide a core signature of item-specific memories in hippocampus.”

- Comparing our results directly to fMRI studies is complex due to vast differences in spatio-temporal resolution and generating mechanisms, and in our opinion, it is outside the scope of the current paper to discuss these various issues in detail. Regardless, our results align conceptually with literature on the representational hierarchy in the ventral stream (Kent, Hvoslef-Eide, Saksida, & Bussey, 2016; Martin, Douglas, Newsome, Man, & Barense, 2018; Moscovitch, Cabeza, Winocur, & Nadel, 2016), some of which is also fMRI-based (Davis et al., 2020; Moscovitch et al., 2016; Prince, Daselaar, & Cabeza, 2005; Saksida, 2009). We discuss this aspect more clearly in the revised version of the manuscript:
 - Discussion, (page 6, paragraph 2): *“Our results support the idea of a representational hierarchy along the ventral visual stream, in which simple visual features are progressively aggregated into more composite features from visual cortex to the hippocampus (Kent et al., 2016; Martin et al., 2018; Moscovitch et al., 2016), with some evidence also coming from fMRI-based studies (Davis et al., 2020; Moscovitch et al., 2016; Prince et al., 2005; Saksida, 2009). However, until now, a direct comparison of the relative importance of early- (simple) and late-layer (composite) features for memory encoding has remained absent from the literature. By using hierarchical vision models, we provide first evidence that the dynamic encoding of composite features in the hippocampus (located at the end of the ventral stream) is substantially more relevant to memory performance than the encoding of simpler features. Thus, our findings provide new and robust evidence for the behavioural importance of the ventral visual representational hierarchy, directly through a crucial new lens of hippocampal neural variability.”*
- Further linking our results to broader theories of memory: We argue just above that our results support the idea of a representational hierarchy during memory encoding, now through the lens of neural variability in hippocampus. However, it is clear that the most visibly missing part of the brain from a representational hierarchy account of memory encoding is the prefrontal cortex (Miller & Cohen, 2001; Moscovitch, 1992). As such, we now include the following passage in Discussion, arguing for how these alternative accounts could relate to a representational hierarchy account:
 - Discussion (page 7, paragraph 5): *“To embed our findings in the context of broader theories of memory encoding, it’s essential to acknowledge the central role the PFC in nearly every classic memory encoding systems account, permitting functions as broad as cognitive control (Miller & Cohen, 2001; Moscovitch, 1992; Ragozzino, 2007) and working memory (Fuster, 1991; Goldman-Rakic, 1996) during memory formation. An interesting alternate account of PFC function during encoding suggests that while the hippocampus may capture dynamic changes in the spatial and temporal aspects of incoming inputs, the (medial) prefrontal cortex may sort those inputs based on their similarity and integrate them over time (Takehara-Nishiuchi, 2020). This function would allow the mPFC to determine the most relevant (and valuable; (Moneta, Garvert, Heekeren, & Schuck, 2023)) content of our encoded experiences (Takehara-Nishiuchi, 2020) an indispensable aspect of effective learning and decision-making. From this perspective, one could predict that PFC would provide sort-and-integrate functionality most efficiently by using aggregated, later-layer stimulus features rather than from simpler features, just as we propose for hippocampus. For example, one could investigate whether the PFC may “sort” inputs by testing whether images with similar late-layer feature content also express similar levels of neural variability. Future work could test these ideas using recordings from both hippocampus and PFC in the context of neural variability during memory encoding.”*

Remark 3

Finally, the major point which is being made appears to be obscured by very vague language and formulation. For example, on page 2, what does the sentence "The processing of different visual features indeed uniquely impacts..." mean? It seems vague.

- **RESPONSE:** We acknowledge that some passages in the paper lacked clarity. We have now carefully revised the paper with a focus on improving clarity. As an example, we now include the new version of the mentioned paragraph below.
 - Introduction (page 1, paragraph 4): *"In recent years, moment-to-moment variability of neural activity has emerged as a behaviourally relevant measure offering substantial insights into brain function beyond conventional approaches such as average brain activity (Waschke, Kloosterman, Obleser, & Garrett, 2021). Indeed, single-neuron spiking rate and variability may support distinct functions in the visual system, as various features of the visual input can affect neural variability independently from mean activity (Hermundstad et al., 2014; Orbán, Berkes, Fiser, & Lengyel, 2016). For instance, computational work suggests that uncertainty with regards to visual input may be uniquely linked to moment-to-moment variability in the spike train (Orbán et al., 2016). Furthermore, a recent study showed that depending on the perceptual statistics of visual input, neurons in V1 of macaques adjust their spiking dynamics (Festa, Aschner, Davila, Kohn, & Coen-Cagli, 2021), leading to trial-by-trial changes in neural variability."*

Interim Summary

In summary, it seems like there are potentially some interesting findings here, but this paper would benefit from a more careful consideration of existing theories of human memory and a more thorough explanation of exactly how the analyses are being performed.

Following are some additional more detailed points which the authors may wish to consider in revision.

- **RESPONSE:** We have taken these comments very seriously. To accommodate the reviewer's feedback, we included a longer discussion of the existing literature in introduction and discussion of the revised manuscript, extended the description of the methods, and finally, made major overall modifications to improve the overall language and description of our analysis pipeline.

Remark 4 (minor)

How do the present results relate to the previously described dependence of neural responses in these areas to low level image properties like brightness and contrast (Steinmetz et al., 2011, Journal of Neurophysiology, 105, 2874-2884). What happens to the relationship to entropy coupling if just the corresponding elements of the current models are considered?

- **RESPONSE:** Thanks for this great point and opportunity for clarification. We expect that global image characteristics like contrast and brightness should in fact globally affect features in the early *and* late model layers. For example, as contrast is a representation of the range of pixel values, if that range approaches zero (e.g., a grey-only background), features will not be detectable at early or late layers, as late layers will not be able to "aggregate" over simpler features that have no content. We thus expect that global image properties such as contrast and brightness should (a) not have any particular corresponding element within our computer vision models *per se*, and (b) should accordingly be reflected in neural variability responses in each region along the ventral visual pathway during encoding. As such, that Steinmetz et al. 2011 found hippocampus to respond to global features such as brightness and contrast makes complete sense in our view.
- Regardless, our results center on the *relative* importance of early and late layers for memory encoding. Each image submitted to our vision models has its own brightness and contrast level. If layer-specific responses differ to that single image, then it cannot be due to the brightness or contrast of that image *per se*. Layer-specific coupling with hippocampal variability is also estimated in our study while within-image brightness and contrast are held constant. As such, we believe our results cannot be directly attributed to brightness or contrast. Although we feel this content might be beyond the scope of our current Discussion section, we would be happy to add points to the Discussion if the reviewer finds them useful.

Remark 5: Rephrase (minor)

p1 para 2 "conjoined" - should this be "conjoined" or even "joined"

Unclear what is meant by the sentence "Beyond the behavioral relevance..."

"We posit that..." This sentence also seems vague because "stronger trial by trial coupling" is ill defined.

- **RESPONSE:** Thanks for highlighting these passages. We have revised them now with more precise phrasing.
 - Introduction, paragraph 2: *"Considering the behavioural relevance of basic sensory encoding in cortex (Pessoa, Gutierrez, Bandettini, & Ungerleider, 2002; Prince et al., 2005), it is plausible that sensory features also need to be directly accessible to the hippocampus to enable the formation of coherent neural representations (Lee, Yeung, & Barense, 2012; Moscovitch et al., 2016). The tracking of simple visual features in the hippocampus may thus play a crucial role to generate rich and detailed memory traces (Behrens et al., 2018; Moscovitch et al., 2016; Yonelinas & Andrew, 2013)."*
 - Introduction, last paragraph: *"Here, we propose the use of partial least squares correlation (PLS) (McIntosh, Bookstein, Haxby, & Grady, 1996) to estimate, for each subject, the relation between (a) trial-by-trial fluctuations of spiking variability in the hippocampus and (b) multivariate stimulus features. We posit that individuals who display stronger trial-by-trial association between hippocampal variability and the content of visual input should also exhibit superior memory, providing evidence that visual features have successfully been encoded by the hippocampus during memory formation."*

Remark 6 (minor)

page 3 - 12 +- 11 hippocampal neurons per subject seems high for well isolated units, though this clearly depends on number of recording channels. Most investigators report perhaps 0.3 - 0.5 well-isolated neurons per channel of recording. It would be good to report average numbers per recording channel and numbers of recording channels here.

- **RESPONSE:** Recordings were performed with macro-micro depth electrodes, each of which contained eight microwires. One microwire functioned as reference, thus allowing the recording of seven microwires per region (Faraut et al., 2018).
- The average number of units recorded (per wire with at least one unit) was 2.3 with a range of 1-7. For further information, please see the original data release note (Faraut et al., 2018) where you can also see a complete histogram showing the number of units isolated for each active wire.

Remark 7 (minor)

p5 "predicted memory formation success over and above the standard spike rate" - how was this demonstrated in any way? It does not appear there was any direct comparison to this alternative model. This is a critical question - does the measure of entropy provide any further information regarding memory formation than firing rate measures?

- **RESPONSE:** Thanks for your comment, we fully agree that this a crucial point for our results. We had already addressed this in the original manuscript, showing that spike rate could not account for our results (see Fig 3b).
- One of our primary results shows that inter-individual differences in the coupling of spiking PE and late-layer image content reflect memory performance. To show that this effect is not driven by spike rate, we ran an additional PLS analysis investigating the latent correlation between spike rate and image content (just as we did for PE). We then examined the partial correlation between spiking PE and memory performance, while controlling for spike rate. The reported partial correlation $\eta_{\text{partial}} = .39$ captures the part of the inter-individual variance in memory performance which was *uniquely* explained by spiking PE. Conversely, spike rate contained no unique prediction of memory performance, when controlling for spiking PE ($\eta_{\text{partial}} = -.14$, $p = .43$).

REVIEWER #2 (Remarks to the Author):

The authors re-analyze an existing dataset of spiking activity recorded in the human medial temporal lobes of epilepsy patients that perform a long-term memory task. In this task, patients are viewing images of objects for encoding. Fifteen to 30 minutes later, the patients are presented with old and new images and are asked to judge whether they saw the images earlier or not, and how confident they are in their memory judgment. Spiking activity is analyzed by computing the trial-wise permutational entropy (PE) of spike trains during encoding. PE is then correlated with stimulus features extracted from early and late layers of artificial neural networks trained to classify images of objects (spatial Sum, SD, and sparsity of the activation patterns in a layer). Correlations of spike PE with early- and late-layer features of the images across trials are computed for each individual patient. These correlation scores are then correlated with a behavioral memory performance score. Late- but not early-layers correlation with PE correlate with behavioral performance. The authors conclude that participants with high coupling of spike-train-variability (spike PE) with late-, and early-layer features profit from more differentiated neuronal activity between stimuli that benefits memory encoding and suggest spike PE as novel measure to analyze neuronal activity.

Overall, I think that the study reads very well and provides an important addition to the literature on the neuronal basis of long-term memory by introducing trial-to-trial spike-train variability in the hippocampus as novel measure that explains inter-individual variance in memory performance.

- **RESPONSE:** We greatly appreciate the positive feedback and would like to thank the reviewers for taking the time to read and comment on our paper.

Remark 1

I have two main points that I think should be addressed. The first one relates to the interpretation of the findings. The authors conclude in the first paragraph of the discussion that “[...] findings of more limited (but still present) behavioral relevance of early-layer features suggest a hierarchy in which the hippocampus downweights simple features without discarding them entirely”. The overall pattern of results, however, seem to me to disagree with the conclusion that spike PE coupling to early-layer features significantly correlates with behavior.

The only result that could support such a conclusion is that individual estimates of coupling of Spike PE with early layer features alone are larger than zero (e.g., Fig. 2b, yellow dots), if I understand correctly. Here, the authors correlate spike PE with early and late features in neural nets on an individual subject level. They find, that both early and late features significantly correlate with spike PE and that correlations of late features with PE are higher than correlations of early features with PE. Furthermore, these correlation scores on the subject level (early features with PE, late features with PE) are correlated with one another across subjects. One question I have here is whether it could be that features of early and late layers are correlated within the neural nets as well. If so, this would probably need to be partialled out. In any case, these correlations do not include behaviour and hence do not directly support the conclusion above.

The correlations of spike PE coupling to early-layer image features with behaviour are all statistically insignificant. The authors nevertheless write “Later memory performance was positively correlated with the coupling of hippocampal spike variability during encoding to image features for early-layers (Fig 3a, $h = .32$, $t_{31} = 1.9$, $p = .07$) [...]” (see also Fig. 3b). If one follows a frequentist approach and assumes $p < .05$ as threshold for significance, there is no significant effect in this case, if I understand correctly. In Supplementary Figure 3, it seems also that measures relating to memory performance (accuracy, d_{prime}) are only positively correlated with late-layer to spike PE coupling.

Most importantly, the in my view most informative analyses in that respect speak against a meaningful relationship of spike PE coupling to early-layer features to behavior. These are the analyses that control for influences of late-layer features that render the association of coupling early-layer features to spike PE with behavior insignificant. Correlations of coupling of late-layer features to spike PE with behavior, in contrast, survive after controlling for coupling of early-layer features to spike PE (Page 5, first paragraph).

- **RESPONSE:** We completely agree with the reviewers' statement that our main finding is the behavioral effect driven by late-layer features. Thus, the emphasis during the discussion should focus on the late layers. We acknowledge that some passages have not been particularly clear in that regard and therefore rephrased these in the discussion (see below).
- As mentioned by the reviewer, we indeed found a significant coupling of spike PE with both early and late layer features (see Fig 3a&b). To show that late-layer features explain memory performance over and above early-layer features, we had already run a regression model partialling out the effect of early-layer features in the original manuscript; the behavioural effects linked to late-layer features remained robust in this control analysis ($r_{\text{partial}} = .45$, $p = .01$, Figure 3b in the main manuscript). Late layer responses are thus essential to explain a substantial part of the across subject variations in memory performance. We now more clearly state this in Discussion.
 - Discussion (page 6, paragraph 2): *“Our results support the idea of a representational hierarchy along the ventral visual stream, in which simple visual features are progressively aggregated into more composite features from visual cortex to the hippocampus (Kent et al., 2016; Martin et al., 2018; Moscovitch et al., 2016), with some evidence also coming from fMRI-based studies (Davis et al., 2020; Moscovitch et al., 2016; Prince et al., 2005; Saksida, 2009). However, until now, a direct comparison of the relative importance of early- (simple) and late-layer (composite) features for memory encoding has remained absent from the literature. By using hierarchical vision models, we provide first evidence that the dynamic encoding of composite features in the hippocampus (located at the end of the ventral stream) is substantially more relevant to memory performance than the encoding of simpler features. Thus, our findings provide new and robust evidence for the behavioural importance of the ventral visual representational hierarchy, directly through a crucial new lens of hippocampal neural variability.”*

Remark 2

Taken together, it seems that the pattern of results fits better with the conclusion that significant coupling of late-layer features to spike PE in the hippocampus is found to correlate with behavior, while correlations of coupling of early-layer features to spike PE with behavior do not survive more conservative analyses controlling for other influences, most importantly the influence of coupling of spike PE with late-layer features. This pattern of results hence supports notions that **solely** representations of more composite stimulus features in hippocampus are behaviorally relevant in my reading. I think this conclusion goes well with the notion that hippocampus operates using an invariant, semantic code as most famously exemplified by the classic findings on concept cells (Quiroga et al. 2004, Nature), and findings where their invariance also extends from the visual in the written and auditory domain as shown in e.g., Quiroga 2009, Current Biology). While these studies do not compare semantic (late layer) vs. more perceptual features (early layers), I contributed to a study where representational similarity of population activity was explained by abstract, categorical features of visual stimuli while low-level visual features of the stimuli could not explain patterns in neural activity (Figure 2 and Supplemental Figure 2 in Reber et al. 2019, PloS Biology).

- **RESPONSE:** We agree that our results tie nicely to the cited studies lending further support to the notion that neural activity in the hippocampus is predominantly driven by abstract features during encoding. Furthermore, we could also establish a strong link to individual differences in behaviour during recognition, revealing that precise encoding of late-layer features predicts memory performance over and above early-layer features.
- We have now revised our discussion to further highlight the connection to existing literature on the neural code of perceptual and semantic features in the hippocampus.
 - Discussion (page 6, paragraph 3): *“Our findings align well with existing literature on single cell and population coding in the human hippocampus showing that hippocampal activity tracks abstract concepts and knowledge (Quiroga, Reddy, Kreiman, Koch, & Fried, 2005; Reber et al., 2019). In our case however, we analysed the visual input continuously with respect to properties of early- and late-layer feature maps instead of using contrasts between objects or categories. Our latent modelling approach then permitted us to leverage trial-to-trial variations of different visual input features, independently of formal semantic image categories, a flexible approach for estimating the encodable content of stimuli without making assumptions about what features might be relevant for encoding the images into memory.”*

Remark 3

The other major point relates to comparing intra- vs. interindividual approaches to analyze their data. Page 1, bottom, the authors write “We posit that individuals with stronger trial-by-trial coupling between hippocampal variability and the content of visual input should also exhibit superior memory, providing evidence that visual features have successfully been encoded by the hippocampus during memory formation.” Focusing on interindividual variance in spike PE across participants to explain interindividual variance in performance is perfectly fine and interesting in my opinion. To get a more complete picture, however, I suggest conducting subsequent memory analyses and compare coupling of spiking PE with features of neural nets during presentation of later remembered vs. forgotten items, as well as those remembered with high vs. low confidence within participant. This would give us an idea of whether the reported phenomenon is indeed something trait like and/or whether it fluctuates also within participant to explain intraindividual differences in performance.

- **RESPONSE:** Thank you for your comment; it's a very interesting point that you are making, but we are afraid it is not feasible to estimate what you mention in the current data. The subject-level PLS estimates the coupling of spiking permutation entropy and image content across trials during the encoding phase. Thus, to ensure robust latent correlations, a fairly high trial count per subject is needed. Of the images presented during encoding, only half were presented again during the recognition phase. Thus, conditioning our analysis on images which were presented during encoding and recognition would eventually cut our trial count in half (from 100 to 50). A further division into remembered vs. forgotten images (high vs. low confidence) would result in a total number of e.g., 40 remembered trials and 10 forgotten trials for an exemplary subject with an accuracy of 80%. Unfortunately, these trial counts are not sufficient to robustly estimate the latent correlation across trials. Furthermore, the across-subject asymmetry with regards to the number of remembered vs. forgotten (or high vs. low confidence) trials would lead to more accurate estimates of remembered images than forgotten ones, which would substantially distort the result of this analysis. Fig S3 in the supplementals illustrates the accuracy (mean = .73) and the average confidence (mean = 2.5) ratings across subjects, which demonstrates that this issue would affect most subjects. Due to these methodological issues, the reviewer's proposed analysis would require an experimental design with a substantially larger trial count. Unfortunately, such analyses are simply not feasible with the given data set.

Supplementals; Figure S3:
Accuracy and confidence in
memory recognition

- However, we agree that it is crucial to illustrate that the behavioural results cannot be attributed to trait-like features in general. Therefore, we ran a control analysis (Fig R4) including the average (perhaps “trait-like” measure of) spiking PE as a covariate in the regression of memory performance on latent coupling between spiking PE and late-layer features. Fig R4 shows that the behavioural

effect remains stable ($\eta_{\text{partial}}=.554$), underlining the importance of adaptive modulations of spiking variability during memory encoding, over and above a “trait-like” signature of average PE.

Figure R4: Control analysis with average PE, Association between memory performance and spiking PE coupling with late-layers after controlling for average spiking PE

Remark 3 (minor)

The authors use a principal component score of various behavioral measures such as accuracy, d-prime and confidence. These measures, however, measure distinct concepts and I do not see the advantage of a composite score. This information is provided in given in Figure S3. There, the reader can appreciate that both accuracy and dprime correlate with late layers but not with early layers, while it seems to be more the case for confidence to be significantly correlated with early- and late-layers couplings to spike PE. I think this should go in the main text and figures.

- **RESPONSE:** Thanks for your comment; we chose to report our results using a composite score across behavioural measures to make the presentation of our findings as concise as possible; we’d refrain from moving individual behavioural measures to the main section, as doing so would vastly increase the number of control analysis and figures included in the main section.
- The individual behavioural measures are reported in detail in the supplemental material, so that the interested reader should be able find them easily.
- We furthermore report the standardized loading of the behavioral measures on the principal component in the methods section (accuracy = .82, confidence weighted accuracy = .92, dprime = .92, and confidence = .53).

Remark 4 (minor)

In Figure 3B and Figure 3C, rightmost panels look liked the data going in the plots are identical, but the stats reported on the figure differ.

- **RESPONSE:** Thanks for pointing this out, we double checked this and there was an error here. We exchanged the figure in 3C with the correct one. Please note however that the reported statistics remained unchanged.

REVIEWER #3 (Remarks to the Author):

Waschke et al. claims that the more precisely hippocampal spiking variability tracks the composite features that comprise each individual stimulus, the better those stimuli are later remembered. The topic of this research is significant to the field of neuroscience as it aims to develop a linking model between sensory encoding and memory formation.

However, to support this claim the authors need to do the following:

1. Reasonably operationalize what they mean by composite features of individual stimulus
2. Reasonably quantify spike variability and operationalize what they mean by “tracking” the entity defined in 1.
3. Reasonably quantify “remembrance” of stimuli and link that to 2.

Overall, while the authors have provided attempts to address all 3 of these aspects, I have struggled to fully check off these points while reading the paper. Therefore, while the overall topic is very interesting, the evaluation of this manuscript, from my end, remains incomplete due to a significant lack of clarity to motivate and demonstrate the evidence supporting the claims. A major revision should be able to make the manuscript clearer and allow for better re-evaluation.

- **RESPONSE:** Thank you for your feedback and taking the time to review our paper. We acknowledge that some sections of the manuscript lacked clarity. We now substantially modified the original version, adding a more extensive explanation of our initial hypothesis, methods, and results (for example, see introduction last paragraph, discussion paragraph 2). Please also see our response to your Remark 3, which comprises a detailed explanation of the methods including the relevant sections from the revised manuscript.

Major comments

Remark 1

1. Are the authors claiming that hippocampal spike variability is partially explained by image stimulus features (in other words ventral stream activity drives the spiking variability in the hippocampus)? And the better it is (at a subject level), the higher performing the subjects are in a memory task? If so, I didn't find any evidence of eye tracking done in the study during the initial encoding phase. For instance, if some subjects simply didn't look at the stimuli (time to time), they will demonstrate a lower link between image features and hippocampal activity (whichever way it is summarized) and a lower performance in the memory tasks. In the absence of eye tracking data, it is important to show that the performance of the participants in the animacy judgement task (during encoding) do not vary significantly and do not correlate with the memory task performance (even though the confound still remains to some degree).

- **RESPONSE:** As the reviewer correctly summarizes, we claim that features of the visual input affect spiking dynamics in the hippocampus. Across individuals, we show that stronger coupling between spiking variability and image features is linked to a better performance during a subsequent memory task. These results suggest that the adaptive modulation of spiking variability plays a key role in the precise memory encoding of perceptual input.
- It would have been great to support our findings with additional eye-tracking data, but unfortunately, single-cell recordings of patients are rare, and eye-tracking wasn't available for the data set we analysed here. Nevertheless, as pointed out by the reviewer, the animacy judgments provide a useful control measure to verify that the patients paid attention during memory encoding (i.e., patients who didn't properly look at the stimuli during memory encoding would show low performance in the animacy judgment task). Furthermore, a correlation between performance in the animacy and memory task would confound the behavioral effect of spike PE modulation, perhaps suggesting that low performing subjects were simply disengaged during memory encoding. We found this to not be the case:
 - On the animacy task, the mean accuracy is 0.98 with a standard deviation of 0.03, illustrating that patients were highly engaged in the task (Fig S4, left). More importantly, performance during the animacy task also does not significantly correlate with recognition memory performance ($r = .1$, $p = .56$, Fig S4, right). Therefore, animacy judgments appear

to have no bearing on our main results. We thank the reviewers for the valuable feedback and have also now included Fig S4 in the supplemental material.

Remark 2

2. Related to the above point, one critical information about the memory task performance is the reliability of the behavioral data per subject. Was there an independent measure of performance reliability? A subject with unreliable data, typically demonstrate lower performance and lower correlation with any entity that is trying to predict that performance.

- **RESPONSE:** We agree that for any behavioural measure, its reliability is essential to ensure accuracy of results. In that regard, the new vs. old recognition task we utilized is a very well-established and widely used paradigm in memory research (Eichenbaum, Yonelinas, & Ranganath, 2007; Rutishauser et al., 2015).
- However, if we understand the reviewer correctly, their main concern is that low performing individuals might show more fluctuations of their performance between successive memory tests. To address this issue, we estimated split-half reliability, which demonstrates the relation between the performance in the first and second half of the recognition test, capturing potential behavioural fluctuations between the two halves (Fig R5). The corresponding split-half reliability coefficient was quite high ($r_{spearman-brown} = .76$).
- It's important to note also that lower performing individuals did not show a larger deviation (i.e., noisier or less reliable) from the regression line than higher performing individuals (Fig R5). This suggests that individual performance level and reliability were unrelated in our dataset; therefore, we are confident that our original analysis was not biased by more unreliable data from lower performers.

Figure R5: Split-half reliability of memory performance

Remark 3

3. The paper is somewhat poorly written, in the sense that the key concepts (especially with relation to the two randomly chosen models HMAX and VGG, and the quantification of feature statistics) are not at all explained and motivated clearly. As a reader I struggled to understand how they relate undefined terms like “composite features that comprise each individual stimulus” to network activation maps, and the first PC etc. It remains so cryptic that a genuine review of that claim isn’t feasible. It is not hard to establish that semantically meaningful constructs can be better linearly separated from features of later layers of CNNs (compared to its earlier layers). If this is the phenomena in the model that the authors are looking to couple with the hippocampal neural spike variability, then there might be more direct ways of establishing it.

- **RESPONSE:** Thank you for your feedback. We acknowledge that a more detailed explanation of our methods is needed. We have included here the revised passages on our criteria to select vision models as well as a longer explanation of the image measures.
 - Introduction (page 1, paragraph 3). This passage from the introduction defines simple and composite features: *“Specifically, feature maps from early layers of these models mark the presence of simple features in the image, while feature maps in later layers resulting from many non-linear combinations of the previous layers mark the presence of more complex, composite features (Riesenhuber & Poggio, 1999; Simonyan & Zisserman, 2015). Hierarchical vision models thus offer a unified framework to characterize both low- and high-level visual features, which may be leveraged by the hippocampus during memory formation.”*
 - Results (page 2, paragraph 2). This revised passage from the main section regards the choice of the two computational vision models: *“To estimate simple and composite visual features of the presented images at encoding, we employed two computational vision models, HMAX and VGG16 (Fig 1B) (Riesenhuber & Poggio, 1999; Simonyan & Zisserman, 2015). The hierarchical structure of these two models allows the differential analysis of simple and composite visual features by means of their layer-wise feature maps (i.e. heatmaps, Fig 1B). Feature maps of early layers capture the presence of simple visual features in the image. These simpler features are then aggregated in a non-linear manner across model layers, resulting in more complex, composite features in the late layers. Importantly, we used estimates from HMAX and VGG16 to counterbalance their respective limitations: HMAX is a biologically inspired model of early visual processing (V1-V4) (Riesenhuber & Poggio, 1999; Serre, Wolf, Bileschi, Riesenhuber, & Poggio, 2007), but is limited in its ability to extract higher-level visual features. Conversely, although not biologically inspired per se, VGG16 is one of the most employed computer vision models in the field; it comprises many more layers than HMAX and is thus better suited for the analysis of more aggregated, higher-level features (Simonyan & Zisserman, 2015).”*
- To further explain the derivation of the image statistics/metrics, it's important to consider the architecture of hierarchical vision models. In these models, each layer comprises multiple “neurons,” with each neuron capturing a specific image feature. In early layers, these features correspond to simple visual patterns. Progressing through the layers, the simple features are non-linearly combined, resulting in more complex features in the later layers. We refer to these as “composite features” in the manuscript. For each image, the vision models generate “feature maps” (heat maps) indicating the location and strength of features in the original image. Based on these within-image feature maps, we calculated three summary metrics for each image: The spatial sum, the standard deviation, and the number of non-zero entries across values in the map.
 - Importantly, the number of features varies by layer depth in the models we used, with later layers containing more features (e.g., VGG layer 1 has 64 features, while layer 5 has 512 features). Thus, to ensure maximal comparability across model types and layers, we did the following (separately) for each model layer and image feature metric (spatial sum, standard deviation, number of non-zero entries): We entered the metrics from all feature maps and all images to PCA, and extracted the first principal component score (this, in effect, puts all images on the same scale for further analysis). The resulting component scores for each model layer and image statistic were then used in the subsequent PLS analysis linking them to spiking PE (see Figure 2).

- We now extended the result and method sections explaining these various aspects in greater detail.
 - Methods, *Extraction of image features from layer-wise feature maps (page 9, paragraph 1&2)* : “ We estimated image features by extracting three feature summary metrics from each model layer and image: (1) the spatial sum and (2) spatial standard deviation (SD) from C1 & C2 layers (HMAX) and from max-pooling layers 1-5 (VGG16), as well as (3) the number of zero elements (“pixels”) for VGG16 max-pooling layers 1-5 (*note that this latter feature metric was not estimable from HMAX due to a near complete lack of zero elements from its layer-wise model output for images used in our stimulus set*). These three feature metrics were chosen to arrive at a comprehensive approximation of image features that incorporates overall saliency (spatial sum), the distribution of salient and non-salient image locations (spatial SD), and the sparsity of saliency maps (number of non-zero entries). Additionally, note that the spatial sum and SD were only computed across non-zero map elements. Importantly, the number of features varies across the layers of HMAX (C1 = 32, C2 = 400) and VGG16 (layers 1-5: 64, 128, 256, 512, and 512, respectively) models. *To ensure maximal comparability across layers, we did the following, separately, for each model layer and image statistic (spatial sum, standard deviation, number of non-zero entries): We entered the metrics from all feature maps and all images to PCA, and extracted the first principal component score (this, in effect, puts all images on the same scale for further analysis)*. Thus, each image participants saw at encoding was represented by three principal component scores for each model layer of interest, one each capturing the layer-wise spatial sum, standard deviation, and number of non-zero entries. These scores subsequently served as input for individual PLS models to estimate the coupling between image features and spiking variability.”
- It’s important to note here that our focus was on the impact of the image-specific feature metrics (as defined above) on neural activity in the hippocampus. This point is rather different from claiming that neural activity is coupled to semantic constructs, as would be the notion of concept cells for example. For this reason, we did not work with contrasts between objects or categories, but instead analysed images on a continuous scale with respect to their early and late layer model features.

Remark 4

4. Establishing whether hippocampal spike variability is better driven by early vs. late layer features of the models can also be done and demonstrated with more standard RSA or linear prediction type methods. Correct neural ceilings need to be estimated and investigated for these types of comparisons to accurately depict how strong these coupling strengths are. More detailed information of how choices in the number of components used from the PCA analysis per layer etc. affect the results is critical.

- **RESPONSE:** Thanks for your comment. We structured the following response by firstly discussing the reasons for using PLS instead of RSA (or related methods) and secondly explaining the choice regarding the number of included PCA components.
- First, standard methods like RSA focus on the similarity between objects or categories and its neural coding. This however was not the research question of the current study; we were instead interested in the impact of *trial-to-trial variations of the image content* (spatial sum, standard deviation and number of non-zero entries of the layer-wise feature maps) on spiking variability in the hippocampus. For this reason, we analyse the covariation between image content and spiking dynamics on a trial-by-trial level, within-subject. Here, joint dimensionality reduction of this covariation (using partial least squares, PLS) is the most fitting method. PLS estimates the maximal association between stimulus features and neural data across trials by projecting them into a joint latent space and extracting the dimensions that reveal maximal multivariate associations. This sensitivity is a major advantage of PLS in comparison to RSA for the analysis of trial-by-trial variations in single cell recordings considering the noisiness of these data. PLS is thus a novel approach in the field of single cell recording, which provides – in the ways applied in the current manuscript – a measure of the continuous coupling between sensory input and neural activity. Such coupling cannot be estimated using RSA or related techniques. Importantly, our main results (see Figure 3) concern the link between subject-wise latent couplings and inter-individual differences in memory performance; for this reason, estimating neural ceilings is not necessary in our view.

- Second, concerning the number of PCA components, it is important to realize that every single layer in a convolutional neural network comprises a multitude of features leading to feature maps of very high dimensionality for each image. Thus, we argue it is *necessary* to reduce the dimensionality of the layer-wise features before combining image data and spiking PE in the PLS; here, the first principal component is the clearest and simplest choice.
 - Including more PCA components by setting a specific level of explained variance per layer is not feasible because the number of features (i.e., the dimensionality) increases from early to late layers; as such, to capture the same amount of variance in across all layers, more PCA components have to be included for late than early layers. This imbalance would distort the results of the subsequent PLS, as the magnitude of latent correlations can greatly benefit from the dimensionality of the input matrices (i.e., including more variables can increase the probability of higher latent correlations because the SVD has more degrees of freedom to work with).
 - Importantly, by taking only the first principal component, we lose proportionally more variance in late layers compared to early layers; for this reason, it is even more striking to us that the late layer model is *most strongly associated* to behaviour.

Remark 5

5. As far as I can tell, no alternate hypothesis of hippocampal activity statistics have been tested. For example, if the analysis is repeated with overall activity levels per trial instead of PE per trial, do we get similar results? I think some analysis of this sort is critical to claim that spike variability is the only key mechanism at play. The authors do test whether controlling for trial-level spike rate change the link between memory performance and the coupling between late layer features and spike variability (Fig 3B, mid panel), but they should also directly report the coupling between trial-level spike rate and late layer features (e.g. the y-axis of Fig 2B, but spike PE replaced by spike Rates).

- **RESPONSE:** Thanks for your feedback. As requested, with Fig R6 we now provide the coupling between trial-level spike rate and late layer features across all subjects (i.e., like Fig 2B, but spike PE replaced by spike rates). The mean latent correlation between spike rate and later layer feature content is $m_{lv_corr} = .4$.
- However, the primary results of our study are *not* about the absolute strength of the latent coupling between spiking variability and image feature in comparison to spike rate, but the link to inter-individual differences in behaviour. Here, the coupling between spike rate and sensory content did not predict memory performance, after regressing out spike PE modulation ($n_{partial} = -.14$, $p = .43$, Fig R6). Conversely, in the original manuscript, we have already shown that the association of spike PE modulation and behaviour remained robust ($n_{partial} = .39$, $p = .02$, see main manuscript Fig 3B, mid panel) after partialling out the coupling between spike rate and image content. For this reason, we concluded that the modulation of spike variability by image content explains inter-individual differences in memory performance over and above spike rate.

Figure R6: Control analysis for spike rate effects. (a) Coupling between late-layer features and spike number across subjects. **(b)** Behavioral effect of spike number modulation on memory performance, after controlling for spike PE.

References

- Behrens, T. E. J., Muller, T. H., Whittington, J. C. R., Mark, S., Baram, A. B., Stachenfeld, K. L., & Kurth-Nelson, Z. (2018). What Is a Cognitive Map? Organizing Knowledge for Flexible Behavior. *Neuron*, *100*(2), 490–509. doi: 10.1016/j.neuron.2018.10.002
- Davis, S. W., Geib, B. R., Wing, E. A., Wang, W.-C., Hovhannisyan, M., Monge, Z. A., & Cabeza, R. (2020). Visual and Semantic Representations Predict Subsequent Memory in Perceptual and Conceptual Memory Tests. *Cerebral Cortex*, *31*(2), 974–992. doi: 10.1093/cercor/bhaa269
- Eichenbaum, H., Yonelinas, A. P., & Ranganath, C. (2007). The Medial Temporal Lobe and Recognition Memory. *Annual Review of Neuroscience*, *30*(1), 123–152. doi: 10.1146/annurev.neuro.30.051606.094328
- Faraut, M. C. M., Carlson, A. A., Sullivan, S., Tudusciuc, O., Ross, I., Reed, C. M., ... Rutishauser, U. (2018). Dataset of human medial temporal lobe single neuron activity during declarative memory encoding and recognition. *Scientific Data*, *5*, 180010. doi: 10.1038/sdata.2018.10
- Festa, D., Aschner, A., Davila, A., Kohn, A., & Coen-Cagli, R. (2021). Neuronal Variability Reflects Probabilistic Inference Tuned to Natural Image Statistics. *Nature Communications*, *12*(1), 3635. doi: 10.1038/s41467-021-23838-x
- Fuster, J. M. (1991). The prefrontal cortex and its relation to behavior. *Progress in Brain Research*, *87*, 201–211. doi: 10.1016/s0079-6123(08)63053-8
- Goldman-Rakic, P. S. (1996). The prefrontal landscape: implications of functional architecture for understanding human mentation and the central executive. *Philosophical Transactions of the Royal Society of London. Series B: Biological Sciences*, *351*(1346), 1445–1453. doi: 10.1098/rstb.1996.0129
- Hermundstad, A. M., Briguglio, J. J., Conte, M. M., Victor, J. D., Balasubramanian, V., & Tkáčik, G. (2014). Variance Predicts Salience in Central Sensory Processing. *ELife*, *3*, e03722. doi: 10.7554/elife.03722
- Kent, B. A., Hvoslef-Eide, M., Saksida, L. M., & Bussey, T. J. (2016). The Representational–Hierarchical View of Pattern Separation: Not Just Hippocampus, Not Just Space, Not Just Memory? *Neurobiology of Learning and Memory*, *129*, 99–106. doi: 10.1016/j.nlm.2016.01.006
- Kriegeskorte, N., Simmons, W. K., Bellgowan, P. S. F., & Baker, C. I. (2009). Circular Analysis in Systems Neuroscience: The Dangers of Double Dipping. *Nature Neuroscience*, *12*(5), 535–540. doi: 10.1038/nn.2303
- Lee, A. C. H., Yeung, L.-K., & Barense, M. D. (2012). The Hippocampus and Visual Perception. *Frontiers in Human Neuroscience*, *6*, 17414. doi: 10.3389/fnhum.2012.00091
- Martin, C. B., Douglas, D., Newsome, R. N., Man, L. L., & Barense, M. D. (2018). Integrative and distinctive coding of visual and conceptual object features in the ventral visual stream. *ELife*, *7*, e31873. doi: 10.7554/elife.31873
- McIntosh, A. R., Bookstein, F. L., Haxby, J. V., & Grady, C. L. (1996). Spatial Pattern Analysis of Functional Brain Images Using Partial Least Squares. *NeuroImage*, *3*(3), 143–157. doi: 10.1006/nimg.1996.0016

- Miller, E. K., & Cohen, J. D. (2001). An Integrative Theory of Prefrontal Cortex Function. *Annual Review of Neuroscience*, *24*(1), 167–202. doi: 10.1146/annurev.neuro.24.1.167
- Moneta, N., Garvert, M. M., Heekeren, H. R., & Schuck, N. W. (2023). Task state representations in vmPFC mediate relevant and irrelevant value signals and their behavioral influence. *Nature Communications*, *14*(1), 3156. doi: 10.1038/s41467-023-38709-w
- Moscovitch, M. (1992). Memory and Working-with-Memory: A Component Process Model Based on Modules and Central Systems. *Journal of Cognitive Neuroscience*, *4*(3), 257–267. doi: 10.1162/jocn.1992.4.3.257
- Moscovitch, M., Cabeza, R., Winocur, G., & Nadel, L. (2016). Episodic Memory and Beyond: The Hippocampus and Neocortex in Transformation. *Annual Review of Psychology*, *67*(1), 105–134. doi: 10.1146/annurev-psych-113011-143733
- Orbán, G., Berkes, P., Fiser, J., & Lengyel, M. (2016). Neural Variability and Sampling-Based Probabilistic Representations in the Visual Cortex. *Neuron*, *92*(2), 530–543. doi: 10.1016/j.neuron.2016.09.038
- Pessoa, L., Gutierrez, E., Bandettini, P. A., & Ungerleider, L. G. (2002). Neural Correlates of Visual Working Memory fMRI Amplitude Predicts Task Performance. *Neuron*, *35*(5), 975–987. doi: 10.1016/s0896-6273(02)00817-6
- Price, & Joseph, L. (2003). Comparative Aspects of Amygdala Connectivity. *Annals of the New York Academy of Sciences*, *985*(1), 50–58. doi: 10.1111/j.1749-6632.2003.tb07070.x
- Prince, S. E., Daselaar, S. M., & Cabeza, R. (2005). Neural Correlates of Relational Memory: Successful Encoding and Retrieval of Semantic and Perceptual Associations. *Journal of Neuroscience*, *25*(5), 1203–1210. doi: 10.1523/jneurosci.2540-04.2005
- Quiroga, R. Q., Reddy, L., Kreiman, G., Koch, C., & Fried, I. (2005). Invariant Visual Representation by Single Neurons in the Human Brain. *Nature*, *435*(7045), 1102–1107. doi: 10.1038/nature03687
- Ragozzino, M. E. (2007). The Contribution of the Medial Prefrontal Cortex, Orbitofrontal Cortex, and Dorsomedial Striatum to Behavioral Flexibility. *Annals of the New York Academy of Sciences*, *1121*(1), 355–375. doi: 10.1196/annals.1401.013
- Reber, T. P., Bausch, M., Mackay, S., Boström, J., Elger, C. E., & Mormann, F. (2019). Representation of Abstract Semantic Knowledge in Populations of Human Single Neurons in the Medial Temporal Lobe. *PLOS Biology*, *17*(6), e3000290. doi: 10.1371/journal.pbio.3000290
- Riesenhuber, M., & Poggio, T. (1999). Hierarchical Models of Object Recognition in Cortex. *Nature Neuroscience*, *2*(11), 1019–1025. doi: 10.1038/14819
- Rutishauser, U., Tudusciuc, O., Neumann, D., Mamelak, A. N., Heller, A. C., Ross, I. B., ... Adolphs, R. (2011). Single-Unit Responses Selective for Whole Faces in the Human Amygdala. *Current Biology*, *21*(19), 1654–1660. doi: 10.1016/j.cub.2011.08.035
- Rutishauser, U., Ye, S., Koroma, M., Tudusciuc, O., Ross, I. B., Chung, J. M., & Mamelak, A. N. (2015). Representation of retrieval confidence by single neurons in the human medial temporal lobe. *Nature Neuroscience*, *18*(7), 1041. doi: 10.1038/nn.4041
- Saksida, L. M. (2009). Remembering Outside the Box. *Science*, *325*(5936), 40–41. doi: 10.1126/science.1177156

- Schacter, D. L., & Tulving, E. (1994). *Memory Systems*. MIT Press.
- Serre, T., Wolf, L., Bileschi, S., Riesenhuber, M., & Poggio, T. (2007). Robust Object Recognition with Cortex-Like Mechanisms. *IEEE Transactions on Pattern Analysis and Machine Intelligence*, 29(3), 411–426. doi: 10.1109/tpami.2007.56
- Simonyan, K., & Zisserman, A. (2015). *Very Deep Convolutional Networks for Large-Scale Image Recognition*. doi: 10.48550/arxiv.1409.1556
- Squire, L. R., & Zola-Morgan, S. (1991). The Medial Temporal Lobe Memory System. *Science*, 253(5026), 1380–1386. doi: 10.1126/science.1896849
- Takehara-Nishiuchi, K. (2020). Prefrontal–hippocampal interaction during the encoding of new memories. *Brain and Neuroscience Advances*, 4, 2398212820925580. doi: 10.1177/2398212820925580
- Urgolites, Z. J., Wixted, J. T., Goldinger, S. D., Papesch, M. H., Treiman, D. M., Squire, L. R., & Steinmetz, P. N. (2022). Two kinds of memory signals in neurons of the human hippocampus. *Proceedings of the National Academy of Sciences*, 119(19), e2115128119. doi: 10.1073/pnas.2115128119
- Waschke, L., Kloosterman, N. A., Obleser, J., & Garrett, D. D. (2021). Behavior Needs Neural Variability. *Neuron*, 109(5), 751–766. doi: 10.1016/j.neuron.2021.01.023
- Yonelinas, & Andrew, P. (2013). The Hippocampus Supports High-Resolution Binding in the Service of Perception, Working Memory and Long-Term Memory. *Behavioural Brain Research*, 254, 34–44. doi: 10.1016/j.bbr.2013.05.030

Response to Reviewer 1

1. The authors have expanded the descriptions of their methods so that the analysis pipeline is clearer. Now that this is clearer, I believe that unfortunately this analysis as well as the additional analyses performed show that the results are not statistically significant. This is because when the fraction of neurons having a PE < 0.0001 is increased, the authors report they do not have enough remaining data to achieve a significant result.

- **Response:** This claim is false. Including all neurons in our analysis by removing the neuron threshold of PE > 0 (equivalent to the threshold PE $> .0001$; see previous revision) actually slightly *improves* the association between latent coupling and memory performance (eta = .551, $p = .001$; our original effect after excluding neurons shown in Fig 3a in the revised manuscript: eta = .54, $p < .001$). In fact, there was no basis for this assumption in the first place as we have not reported this particular result previously. We can only assume that the reviewer confused two separate preprocessing steps of our study; the exclusion of silent *neurons* and the exclusion of silent *trials* from the encoding phase.

2. While the authors consider the impact of this pre-exclusion and the potential rationale for doing so, it appears the analysis does not consider the fundamental point of both Kriegeskorte et al 2009 and Thorp and Steinmetz 2007. That is - when you pre-select a set of data based on one statistical test and then use that subset in a subsequent test, you often have biased the distribution of that data so that your estimate of the significance level, or likelihood of having a type-I error, in the subsequent test is grossly incorrect, potentially by up to 1800%. In this case, the authors select neurons based on the PE and then subsequently used a statistic derived from the PE, namely computing the PLS between the PE and image features. The relationship between the PLS and subsequent memory performance is one of the primary claims of this paper.

- **Response:** The claim that we used a statistical test to pre-select neurons is also false. There was no test; instead, we excluded only *completely silent neurons* from further analyses. These neurons displayed no spiking across all encoding trials of the experiment (and thus by mathematical necessity, have a permutation entropy (PE) value = 0; as clarified in our previous response). This is standard procedure during the preprocessing of neurophysiological data, as there can be many potential factors causing such null-data in single-neuron recordings, from technical reasons (e.g., SNR) to phenomenological ones (e.g., the cell is not needed for a given cognitive process). Crucially, the removal of silent cells was done *a priori* and fully agnostically to any particular stimulus, behavioural performance measure, or subsequent analysis.
- **Response:** It is also crucial to note that the general reasoning as put forward in Kriegeskorte et al. 2009 and Steinmetz & Thorp 2013 is *not* applicable to our analyses (also, despite an extensive search, we were not able to find the reviewer's cited Thorp & Steinmetz 2007 paper anywhere online). A key premise in both papers is that data selection and subsequent statistical tests are performed on the same statistical variable. However, this is not the situation we find ourselves in.

The reviewer may have missed that our main outcome statistics are based on two already fully separate data sources (encoding vs. recognition trials). Specifically, the main result of

our study is the between-subject regression of memory performance on image-neuron coupling as shown in Fig 3 of the manuscript; here, memory performance was estimated *only* from recognition trials, while image-neuron coupling estimates were taken *only* from encoding trials. Encoding and recognition trials represent two naturally and fully dissociated experimental sessions, with a 30-minute break in between (see Methods).

Since encoding and recognition trials are already separate in our design, the exclusion of silent trials only from the encoding phase cannot lead to an “overestimation of the significance level” of an analysis linking to performance on recognition trials (and, note that no trials were removed from the recognition phase). Crucially, as should have been clear in the past revision, the *estimation of coupling between PE and image features is not at all informed by recognition performance itself*; the link to recognition performance occurs *only after* encoding-based coupling between PE and image features has been estimated for each subject individually.

Furthermore, the reviewer does not seem to appreciate that we have already controlled for the number of trials and neurons included per subject (Fig 3b in the manuscript) to offset the concern that the extent of thresholding during the encoding phase matters for our primary associations with memory performance during the recognition phase. The effect size in this control analysis ($\eta = 0.64$) was even stronger than the initial regression without covariates. Thus, our primary results cannot at all be explained by the extent to which neurons or trials were excluded, further supporting the robustness of our conclusions.

3. There are several ways in which this issue can normally be addressed. One is to choose neurons based on the PE for one randomly selected subset of trials and then compute the derived statistic based on the other trials. The report of a similar analysis in the response to reviews suggests this will lose so much power that a significant result will not be obtained, though the authors do report such a test directly. The other approach is to apply either the same or a different statistic which perhaps represents more directly what the authors wish to measure to *all neurons on all trials* and see if a significant result is obtained. Again the description of the subsetting reported in the response strongly suggests that this will not produce a significant result. The results are certainly interesting and this sort of pre-selection is actually rather common in neuroscience. However, without actual demonstration of a significant result, this reviewer cannot recommend publication.

- **Response:** We have made various arguments above against circularity in our study and extensively outlined our rationale in the last revision for why keeping all null *trial* data is neither experimentally nor statistically sensible in our study. Regardless, perhaps the most straightforward approach to now conclusively rule out the reviewer’s concern that *any* thresholding we have applied (whether trial or neuron) could have impacted our primary results is to recompute all key analyses using non-parametric permutation tests (while thresholding and statistical power are *identical* to our original analyses). The reasoning behind the use of permutation tests in this situation is that *any* potential overinflation/bias the reviewer believes to be introduced by thresholding should also be present when the data are randomized; in this case, the recovered effect size would not be related to a true underlying effect, but would be explained by the thresholding itself (see Steinmetz & Thorp, 2013). Such permutation tests thus provide an empirical (rather than statistically assumed) null distribution to which our primary results can be compared. If the reviewer’s concerns are accurate, using a permuted null distribution should dramatically limit our ability to detect significant effects in our study due to the presence of bias/circularity. We verify that this is absolutely not the case by doing the following:
 - a) We randomly permuted the trial-wise PE values at encoding (1000x) for each patient and then re-computed (1000x) the subject-wise encoding-based coupling between spike PE and late-layer image features using PLS. The resulting coupling values were then used to rerun (1000x) the between-subject regression of recognition performance on the encoding-based coupling of spike PE and image features, yielding a null distribution of η values (Fig R1a, top panel). In the same manner, we also recomputed the regression with memory

performance, controlling for the number of trials and neurons included per subject (Fig R1b).

- b) We then reran the late layer analysis, controlling for the coupling of spike PE to early layers image features (Fig R1c) and controlling for the coupling of spike rate to late layer image features (Fig R1d). Importantly, for these controls, we re-computed the coupling of spike PE to early-layers and the coupling of spike rate to late-layers using the *same* 1000 permutation runs as for the spike PE to late-layer model to make these immediately comparable. Then we used these estimates as covariates in the final regression with recognition performance, resulting in a distribution of 1000 partial etas.
- c) For all analyses, we computed the permuted p-values as the proportion of permutations revealing a higher eta estimate than our estimate from the original, unpermuted data.

- Our results are noted in Figure R1 below. Here, we show that no p-value resulting from these newly computed permutation tests differs appreciably from our original results, and that *every single one of our previous inferences remain intact*. The additional analyses laid out here provide a clear-cut “demonstration of significant results” requested by the reviewer and should conclusively address their remaining concerns. We have adjusted all p-values in the main manuscript to reflect these newly computed permutation tests and added the description of the permutation tests to the Online Methods (all changes in red font).

Fig. R1: Permutation tests of the association between memory performance and the coupling of spike PE to late-layer image features. Upper panels show the distribution of effect sizes across 1000 permutations, and vertical grey bars indicate the original effect size from our non-permuted data. Lower panels show the original scatters and correlation values noted in Figure 3 of our previously revised manuscript, now with newly computed permuted p-values (in red font). (a) Zero-order estimates for late-layer coupling vs. performance. (b) late-layer coupling vs. performance, controlling for number of trials and neuron, age, task version and image duration. (c) late-layer coupling vs. performance, controlled for early-layer coupling. (d) late-layer coupling vs. performance, controlled for spike rate coupling to late-layers.